# A bacterial defense system targeting modified cytosine of phage genomic DNA

Rui Liu[1,5], Dongmei Tang[2,3,5], Mingze Niu[1], Shikun Lei[1], Zhiyong Zong ®[4] ✉, Qiang Chen ®[1] ✉ & Yamei Yu ®[1] ✉

The evolutionary arms race between bacteria and phages drives the development of bacterial antiviral defense systems and phage counter-defense strategies. Restriction–modification (RM) systems protect bacteria by methylating 'self' DNA and cleaving unmodified phage DNA. Phages like T-even coliphages evade RM systems by substituting cytosine with 5-hydroxymethyl cytosine (5hmC) or 5-glucosylated hmC (5ghmC). Here, we characterize a single-component antiviral defense system featuring a GIY-YIG endonuclease domain. Biochemical and structural analyses demonstrate that this defense system is a type IV modification-dependent restriction endonuclease that specifically degrades 5hmC- or 5ghmC-modified DNA, and we accordingly name it CMoRE (**C**ytosine **Mo**dification **R**estriction **E**ndonuclease). The crystal structures reveal an N-terminal GIY-YIG nuclease domain and a C-terminal modification-sensing domain. Unique features, including a 'GIYxY-YIG' motif and an inhibitory negatively charged loop, distinguish CMoRE as an additional member of the GIY-YIG family. This system not only highlights the evolutionary interplay between phages and bacteria but also presents CMoRE as a potential tool for precise genomic detection of 5hmC in mammals, with implications for epigenetics research and disease diagnostics.

The arms race between bacteria and phages exerts strong evolutionary pressure on both sides. Bacteria have evolved various antiviral defense systems, while phages have developed counter-defense strategies. For example, the well-characterized restriction–modification (RM) systems are by far the most abundant anti-phage systems, present in 83% of sequenced prokaryotic genomes[1]. A typical RM system consists of two enzymatic components: a methyltransferase and a restriction endonuclease. These two enzymes both recognize and act on a certain DNA sequence. The methyltransferase labels the 'self' genomic DNA by methylation while the restriction endonuclease cleaves the unmodified "non-self" DNA, respectively[2]. To resist RM systems, the T-even coliphages substitute their genomic cytosines with 5-hydroxymethyl cytosine (5hmC) which may be further modified into 5-glucosylated hmC (5ghmC)[3].

Modifications at the 5th carbon of cytosine are also present in mammalian genomic DNA. 5-methyl cytosine (5mC) and its oxidation product 5hmC have been referred to as the fifth and sixth DNA bases, respectively[4,5]. The extensively investigated 5mC is a prominent epigenetic modification which plays a well-established role in gene expression, cellular differentiation, genome stability and disease development[6]. Aberrant levels of 5hmC have been implicated in various cancers and neurological disorders, making it a promising epigenetic marker for the diagnosis, treatment and prognosis of related diseases[7,8].

Here, we characterize a newly identified antiviral defense system, PD-T4-3, which is a single-component defense system containing a GIY-YIG endonuclease domain initially found in *E. coli* ECOR68[9]. We demonstrate that it is a type IV modification-dependent restriction

[1]Department of Biotherapy, Cancer Center and State Key Laboratory of Biotherapy, West China Hospital, Sichuan University, Chengdu, China. [2]Department of Urology, State Key Laboratory of Biotherapy, West China Hospital, Sichuan University, Chengdu, China. [3]School of Medicine, University of Electronic Science and Technology of China, Chengdu, China. [4]Center of Infectious Diseases, West China Hospital, Sichuan University, Chengdu, China. [5]These authors contributed equally: Rui Liu, Dongmei Tang. ✉e-mail: zongzhiy@scu.edu.cn; qiang_chen@scu.edu.cn; yamei_yu@scu.edu.cn

endonuclease and specifically degrades DNA modified by 5hmC or 5ghmC, and we thus rename it CMoRE (**C**ytosine **Mo**dification **R**estriction **E**ndonuclease). Crystal structures reveal that CMoRE contains an N-terminal GIY-YIG nuclease domain and a C-terminal modification-sensing domain. CMoRE's nuclease domain is characterized by a "GIYxY-YIG" motif and an inhibitory negatively-charged loop, thus represents an additional member of GIY-YIG family. Our data also suggest that CMoRE is a potential tool to facilitate the accurate detection of 5hmC at the genomic level.

## Results

### CMoRE system rendered *E. coli* defense against T-even coliphages

CMoRE is a single-component defense system which contains a GIY-YIG endonuclease domain and a function-unknown domain which we here prove to be a DNA modification sensing domain (Fig. 1a). We cloned this gene of three strains (*Escherichia coli* APEC O1, *Escherichia coli* O157:H7, and *Klebsiella pneumoniae* 30684/NJST258_2) into the pET28a vector, under the control of the corresponding native promoter, respectively. The anti-phage assays were performed in *E. coli* BL21(DE3) which naturally lacks this defense system. We challenged the system-containing strain with a set of phages. We found that CMoRE system protected *E. coli* against phages T2, T4 and T6, reducing plating efficiency by about 5 orders of magnitude (Fig. 1b and Supplementary Fig. 1). The CMoRE system from *E. coli* APEC O1 showed much weaker antiviral activities compared to the other two. Replacing its native promoter with T7 promoter resulted in a more robust phage resistance (Fig. 1b and Supplementary Fig. 1).

Under high-titer phage infection (MOI > 1), the system-containing cells grew normally compared to the control (Fig. 1c), indicating that CMoRE functioned via a direct defense mechanism instead of abortive infection. The same observation has been reported for this defense system from *E. coli* ECOR68[9].

### CMoRE system specifically degraded DNA containing 5hmC or 5ghmC

It seemed that the CMoRE system displayed an antiviral function specifically against T-even phages (Fig. 1b). It is a feature of T-even phages to contain 5hmC and/or 5ghmC in place of ordinary cytosine in their DNA[3]. Based on these results, we speculate that the CMoRE system specifically recognizes and degrades phage genomic DNA containing modified cytosine.

To test this hypothesis, we expressed and purified CMoRE and evaluated its endonuclease activity against the genomic DNA of various phages. Our results clearly showed that CMoRE specifically degraded the genomic DNA of T-even phages (Fig. 2a and Supplementary Fig. 2a). The endonuclease activity of CMoRE showed a divalent cation preference of $Mg^{2+}$ and $Mn^{2+}$ followed by $Co^{2+}$, $Ca^{2+}$, $Cu^{2+}$ and $Ba^{2+}$, whereas $Zn^{2+}$ and $Ni^{2+}$ did not support its endonuclease activity (Supplementary Fig. 3). Consistently, the quantitative PCR (qPCR) results showed that the T4 phage DNA was significantly suppressed in a very low level with the presence of CMoRE system (Fig. 2b). Then we performed next-generation sequencing of *E. coli* cells infected with T4 phage. The sequencing results showed that the phage DNA was severely depleted across the entire T4 genome with the presence of CMoRE system, suggesting a lack of strict sequence restrictions (Fig. 2c). In contrast, the host genome remained unaltered under these conditions (Fig. 2c). The preservation of host DNA further confirmed that CMoRE could distinguish self from non-self.

We further evaluated the endonuclease activity of CMoRE using PCR-amplified products incorporating various modified cytosines as substrates. While no detectable endonuclease activity was observed against DNA containing ordinary cytosine or 5mC, CMoRE efficiently degraded DNA containing 5hmC or 5ghmC (Fig. 2d and Supplementary Fig. 2b). To assess the substrate specificity of CMoRE in vivo, we employed two T4 phage mutants: deglycosylated T4(hmC) (deleting both α- and β-glucosyltransferase) and unmodified-cytosine T4(C)

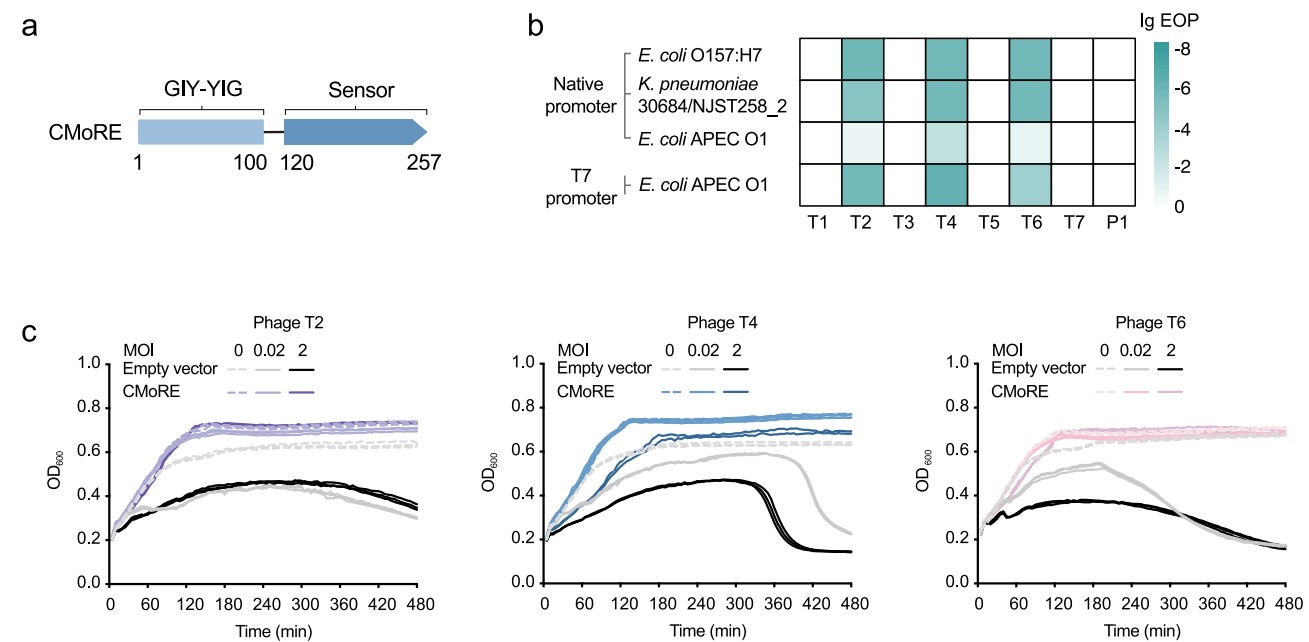

**Fig. 1 | The CMoRE system confers defense against T-even phages. a** Schematic diagram of the CMoRE system. The domain boundaries are annotated based on the *E. coli* O157:H7 CMoRE system. **b** Anti-phage activity of CMoRE systems. Efficiency of plating (EOP) of phages is measured by serial dilution plaque assays with *E. coli* BL21(DE3) strain. Data represent an average of three biological replicates. **c** Phage infection in liquid cultures of the *E. coli* BL21(DE3) strain containing *E. coli* O157:H7 CMoRE system. The strain containing empty vector is used as the control. Cells are infected at various MOI values. For each MOI, results of three experiments are presented as individual curves. Source data are provided with this paper.

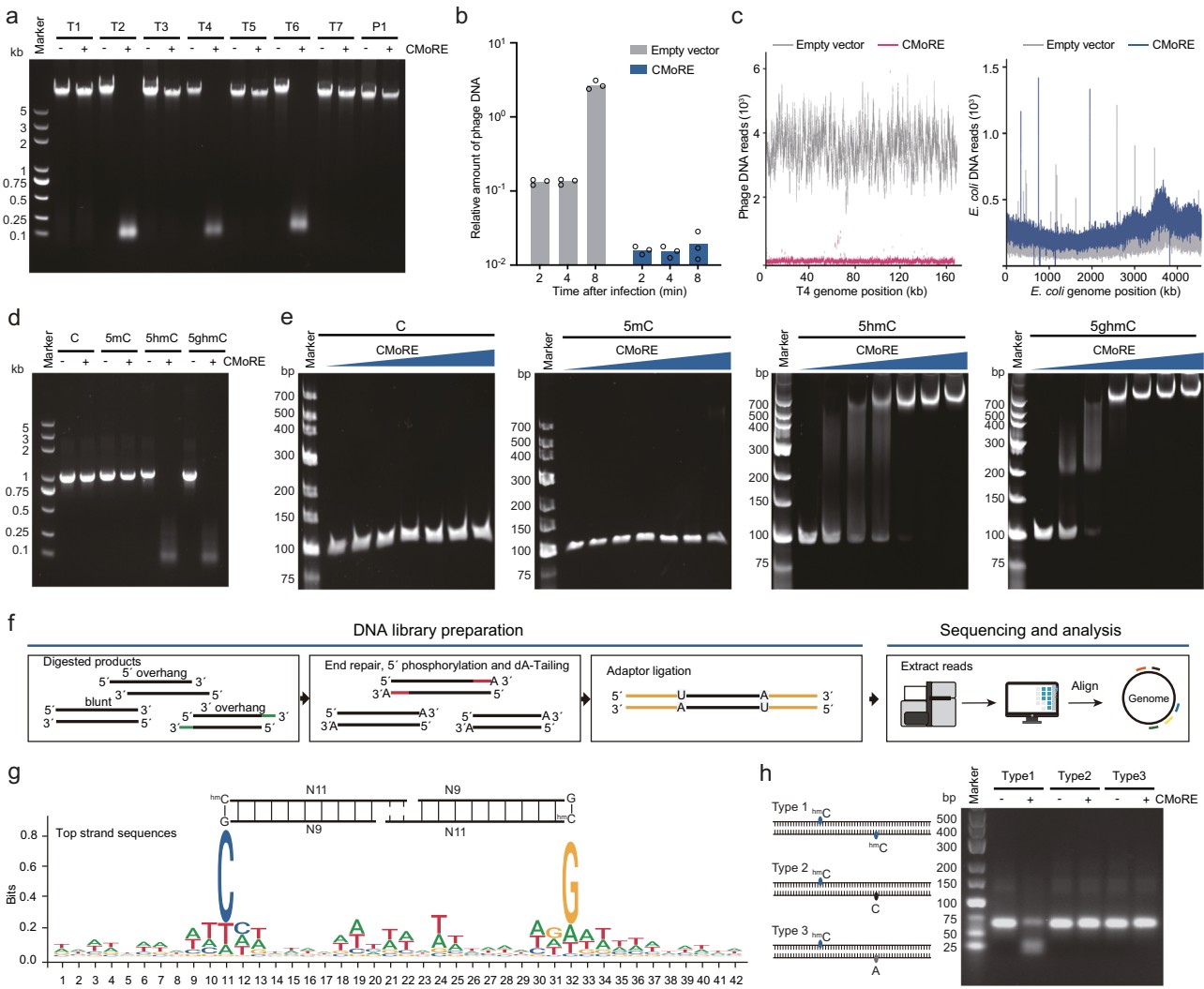

**Fig. 2 | *E. coli* O157:H7 CMoRE specifically degrades 5hmC- or 5ghmC-modified DNA. a** CMoRE degrades the genomic DNA of T-even phages. **b** Quantitative PCR analysis of T4 phage DNA in the presence or absence of the CMoRE system. At each time post infection, phage-specific primers are used for the measurement, with host 16S rRNA serving as an internal normalization control. Bar graphs represent the mean of three biological replicates. **c** Normalized next-generation sequencing reads mapped to the T4 phage and *E. coli* genomes. DNA samples for sequencing are harvested 8 min post-infection. **d** Endonuclease activity of CMoRE against PCR-amplified products with various cytosine modifications. **e** Electrophoretic mobility shift assay of CMoRE binding to PCR-amplified DNA containing unmodified cytosine, 5mC, 5hmC, or 5ghmC. Increasing concentrations of CMoRE proteins (0, 0.1, 0.2, 0.5, 1, 2, or 5 μM) are incubated with 0.1 μM DNA. **f** Schematic diagram of library construction and sequencing workflow for CMoRE-digested DNA fragments. **g** Sequence logo representation of the CMoRE cleavage sites. **h** Endonuclease activity of CMoRE against synthesized modified DNA substrates. Source data are provided with this paper.

(harboring an amber mutation in dCMP hydroxymethylase and a deletion in dCTPase). Plaque-forming assays revealed that T4(hmC) remained susceptible to CMoRE, whereas T4(C) exhibited complete resistance to CMoRE-mediated defense (Supplementary Fig. 4), confirming that cytosine modification is essential for CMoRE mediated defense. These in vitro and in vivo data suggested that CMoRE system fulfilled its anti-phage function via specifically degrading 5hmC- or 5ghmC-modified phage DNA.

To understand the substrate preference of CMoRE endonuclease, we used electrophoretic mobility shift assay (EMSA) to evaluate the binding abilities of CMoRE to different DNA substrates. EMSA results showed that CMoRE bound 5hmC- or 5ghmC-modified DNA but failed to bind ordinary or 5mC-modified DNA (Fig. 2e). The difference of binding affinity provided an explanation for the substrate specificity of CMoRE.

To determine the cleavage pattern of CMoRE, we sequenced the digested products of T4 phage genomic DNA and analyzed the cleavage sites from mapped DNA fragments. The libraries of restriction fragments were generated by end repair, 5′ phosphorylation, dA tailing and adaptor ligation (Fig. 2f). The calculated distance between the mapped DNA fragments is predominantly 2-bp (Supplementary Fig. 5), suggesting the digested products contain 2-nt 3′ overhangs. The restriction fragments revealed a consensus sequence $CN_{11}/N_9G$, and the cleavage occurs at defined locations to the modified base (11 bases on the modified strand, 9 bases on the other) (Fig. 2g). This symmetric pattern of the cleavage site implies that CMoRE may function as dimers. Then, we synthesized DNA substrates with various 5hmC-modified patterns to confirm that CMoRE recognition indeed requires two 5hmC sites (Fig. 2h).

## Structure of CMoRE

To further understand the mechanism of action of CMoRE, we determined the crystal structures of CMoRE from *E. coli* APEC O1 and *E. coli* O157:H7 at resolutions of 2.2 Å and 2.8 Å, respectively. The CMoRE structure adopts a two-domain configuration. An N-terminal GIY-YIG catalytic domain and a C-terminal domain are linked by a flexible loop

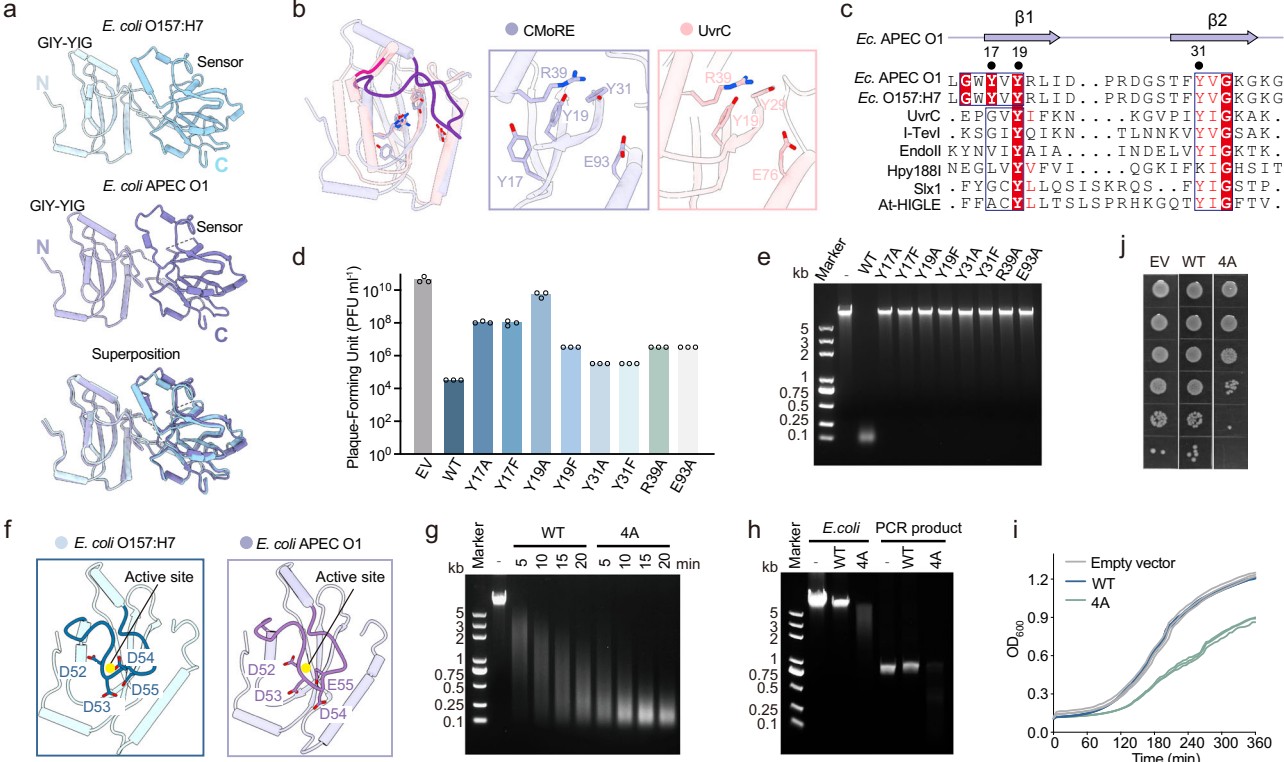

**Fig. 3 | Crystal structure of CMoRE. a** Overall structures of CMoRE from two bacterial species and their structural superposition. **b** Superposition of the GIY-YIG domains from *E. coli* APEC O1 CMoRE and UvrC (PDB ID 1YCZ). The α2-α3 loops are highlighted by dark colors. Details of the active site are shown in the enlarged parts. **c** Structural-based sequence alignment of the GIY-YIG motifs from structure-known GIY-YIG members. **d** Plaque formation assays of T4 phage infecting the *E. coli* BL21(DE3) strain harboring empty vector, CMoRE, or CMoRE mutants. Data represent plaque-forming units (PFU) per milliliter of each phage infection. Bar graphs show the average of three biological replicates with individual data points overlaid. **e** The endonuclease activity of wild type and mutant *E. coli* O157:H7 CMoRE (as in **d**) against T4 phage genomic DNA. **f** The negatively-charged loop blocks the active site. The four consecutive negatively-charged residues are shown and labeled. The active site is indicated by a yellow circle. **g** Endonuclease activity of wild type *E. coli* O157:H7 CMoRE or mutant 4A. T4 phage genomic DNA is used as the substrate. Enzymatic reactions are performed at room temperature. **h** *E. coli* O157:H7 mutant 4A displays endonuclease activity against *E. coli* genomic DNA and PCR product. **i** Growth curves of *E. coli* BL21(DE3) strain expressing wild type *E. coli* O157:H7 CMoRE or the mutant 4A. Empty vector is used as the control. Results of three experiments are presented as individual curves. **j** Colony formation assay demonstrating cellular toxicity of *E. coli* O157:H7 mutant 4A. Source data are provided with this paper.

(Fig. 3a). In both CMoRE structures, the linker between the two domains lacks enough electron density probably due to its inherent flexibility. The CMoRE structures from the two species could be superimposed very well, with a root mean square deviation (RMSD) of 0.93 Å for 225 Cα atoms (Fig. 3a).

The structural comparison in Dali server suggests that the GIY-YIG domain of CMoRE is most similar to another GIY-YIG family member UvrC, which catalyzes the 3′ incision reaction during nucleotide excision repair[10]. Structural superposition showed that the central β-sheet and two flanking α-helixes of these two GIY-YIG domains could be well aligned (Fig. 3b). The catalytic arginine and glutamic acid residues are conserved (Fig. 3b). Notably, between the two conventional tyrosine residues (Y17 and Y31) of the "GIY-YIG" motif, there is a third tyrosine residue (Y19) forming a "GIYxY-YIG" motif (Fig. 3b, c). Structure-based sequence alignment of the structure-known GIY-YIG endonucleases indicates that Y19 of CMoRE, not Y17, corresponds to the tyrosine residue of "GIY" in other GIY-YIG endonucleases (Fig. 3c). We introduced specific point mutations to the key residues in the active site of CMoRE's GIY-YIG domain. Each mutation impaired the anti-phage function of CMoRE (Fig. 3d). All mutations abolished the in vitro endonuclease activity of CMoRE (Fig. 3e). Interestingly, while mutations of the two conventional tyrosine residues (Y17 and Y31) to alanine or phenylalanine reduced anti-phage activity to a similar level, the Y19F mutant retained significantly higher activity than Y19A mutant (Fig. 3d). These results suggest that Y17 and Y31 likely function via their

hydroxyl groups, whereas Y19 may rely on both its hydroxyl group and the hydrophobic aromatic ring for activity.

## Autoinhibition of CMoRE nuclease activity

Compared to UvrC, CMoRE owns a 15-residue inserted loop between α2 and α3 (Fig. 3b). This inserted loop is rich of negatively-charged residues and located over the active site like a lid (Fig. 3f). Since DNA is a highly negatively charged molecule, we speculate that the negatively-charged inserted loop may interfere the substrate binding.

We introduced mutations to the inserted loop to remove the negative charges by substituting the four consecutive negatively-charged residues with alanine, and we herein call it mutant 4A. The purified mutant 4A showed higher endonuclease activity compared to the wild type protein (Fig. 3g). What is more, the mutant 4A displayed endonuclease activity against ordinary DNA (Fig. 3h), although with a much lower efficiency compared to T4 phage DNA (Supplementary Fig. 6). Consistently, bacterial cells expressing the mutant 4A showed a slower growth rate (Fig. 3i), implying cell toxicity. We then confirmed the cytotoxicity of the mutant 4A by plating assays (Fig. 3j). These results suggest that the negatively-charged loop plays an autoinhibitory role by interfering the substrate DNA binding via charge repulsion.

## The tetrameric assembly of CMoRE

In the crystal structure of *E. coli* O157:H7 CMoRE, there are four molecules per asymmetric unit. Two CMoRE molecules form a dimer in

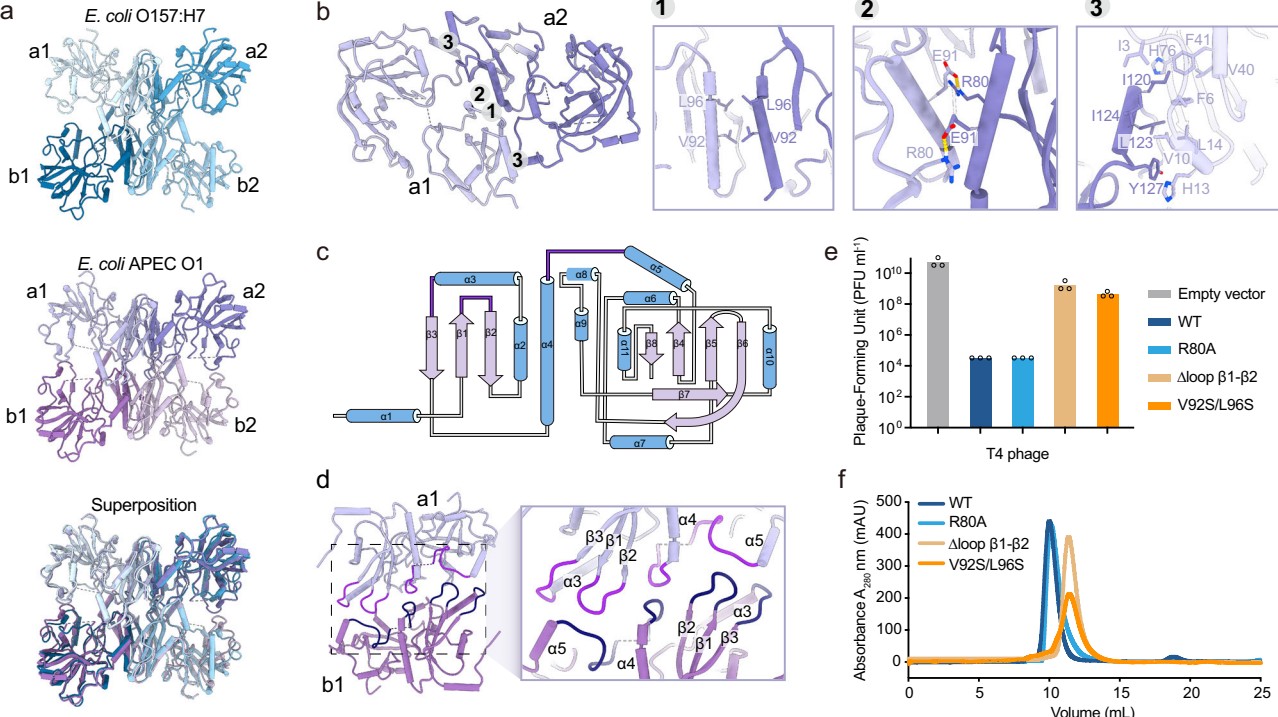

**Fig. 4 | The tetrameric assembly of CMoRE. a** The symmetric dimer of dimer arrangement of CMoRE tetramers from two bacterial species and their super-position. **b** The intra-dimer interface of CMoRE assembly. Details of the 2-fold interface are shown in the enlarged parts. **c** Topology diagram of CMoRE, high-lighting three loops (purple) critical for inter-dimer interactions. **d** The inter-dimer interface of CMoRE assembly. Details of the interface are shown in the enlarged part. **e** Anti-phage activity of wild type and mutant CMoRE against T4 phage. Bar graphs show the average of three biological replicates with individual data points overlaid. **f** Size-exclusion chromatography profiles for the wild type CMoRE and mutants. Source data are provided with this paper.

a head-to-tail arrangement with twofold symmetry, and two dimers form a tetramer in a back-to-back arrangement (Fig. 4a). In the crystal structure of *E. coli* APEC O1 CMoRE, there is only one molecule per asymmetric unit, and four symmetry-related molecules form a homo-tetramer (Fig. 4a). The CMoRE proteins from these two species adopt a nearly same tetrameric assembly (Fig. 4a).

Upon the tetramer formation, ~20% of the total solvent-accessible surface area of CMoRE is buried. The twofold symmetric intra-dimer interface involves α1, α2, α4, α5 and β3, including extensive hydro-phobic interactions and several hydrogen bonds (Fig. 4b, c). The inter-dimer interface is also twofold symmetric, mainly contributed by three loops (Fig. 4c, d). To determine whether the tetrameric assembly is essential for CMoRE's function, we engineered mutations targeting two interfaces: the intra-dimer interface (R80A, V92S/L96S) and the inter-dimer interface (Δloop β1-β2). The R80A mutant retained full anti-phage activity, whereas the V92S/L96S and Δloop β1-β2 mutants exhibited severe impairment of anti-phage activity (Fig. 4e). Size exclusion chromatography analysis showed that the R80A mutant did not disrupt oligomerization, whereas the V92S/L96S and Δloop β1-β2 mutants abolished oligomerization (Fig. 4f). This strong correlation between oligomeric states and anti-phage activities supports that CMoRE's tetrameric architecture is critical for its anti-phage function.

## Structure of CMoRE complexed with modified cytosine
To understand the substrate recognition of CMoRE, we determined the crystal structures of CMoRE complexed with 5hm-dCTP. We co-crystallized CMoRE from *E. coli* APEC O1 or *E. coli* O157:H7 with 5hm-dCTP and obtained the complex structures at resolutions of 2.15 Å and 3.0 Å, respectively. In both structures, the 5hm-dCTP molecules bound to a same position of the C-terminal domain (Fig. 5a, b). The structural data suggest that the C-terminal domain of CMoRE is the substrate recognition domain.

In the complex structure, the 5-hydroxyl group of 5hmC formed a hydrogen bond with the main-chain oxygen of Y127 (Fig. 5a). In the structure of another 5hmC recognition domain, the SET and RING finger-associated (SRA) domain of ubiquitin-like with PHD and RING finger domains 2 (UHRF2), the 5hmC base is flipped out of the DNA duplex to bind UHRF2-SRA[11]. Structural comparison demonstrates that the recognition mode observed in our CMoRE /5hm-dCTP complex closely parallels this DNA-bound conformation (Fig. 5a).

In the crystal structure, the linear distance between the two 5hmC moieties within a CMoRE dimer is ~53 Å, while the linear distances between inter-dimer 5hmC moieties are above 80 Å (Fig. 5c). The distance between the two cytosines in the 5hmC recognition pattern ($CN_{11}/N_9G$) is 21-bp (Fig. 2g), corresponding to about 71 Å. Thus, it is tempting to speculate the two 5hmC sites of the DNA substrate are recognized by a CMoRE dimer, and a CMoRE tetramer could simultaneously bind two DNA substrates.

## Defense mechanism of CMoRE against T-even phages
Together, our results suggest that CMoRE is a type IV modification-dependent restriction endonuclease (MDRE) that specifically targets 5hmC- or 5ghmC-modified DNA. MDREs typically consist of separate modification-sensing and nuclease domains[12–14]. The crystal structures reported here demonstrate that CMoRE contains two separate domains: an N-terminal GIY-YIG nuclease domain for cleavage catalysis and a C-terminal modification-sensing domain for ligand recognition.

One possible activation mechanism is that the recognition of the modified cytosine leads to a conformational change of the auto-inhibitory loop to enable degradation of phage DNA. However, the structural comparisons between apo and ligand-bound states demonstrated subtle conformational alterations in both the auto-inhibitory loop and overall tetrameric assembly (Supplementary Fig. 7). To investigate whether CMoRE employs an allosteric

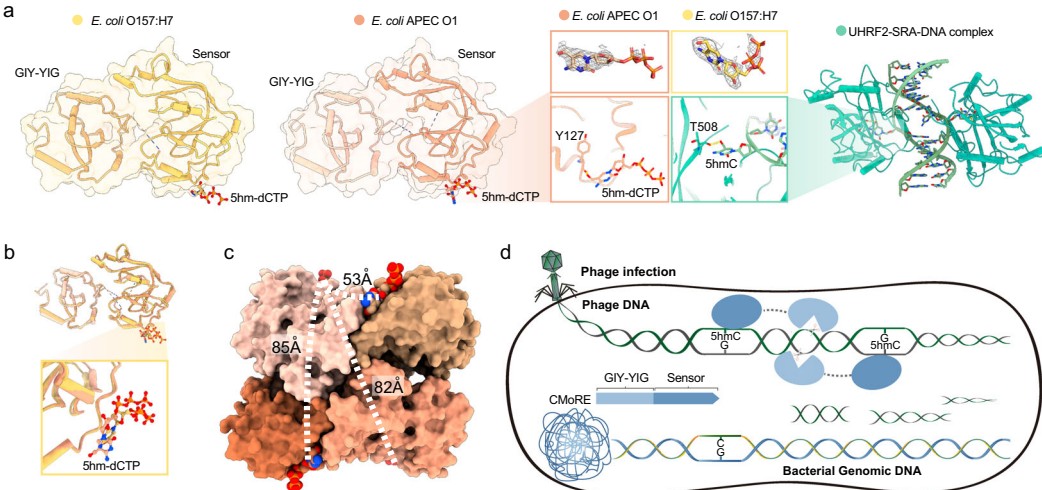

**Fig. 5 | CMoRE recognizes 5hmC. a** Crystal structures of CMoRE complexed with 5hm-dCTP. The electron density maps of 5hm-dCTP (Fo-Fc omit map, contoured at 3 σ) are shown. The details of the 5hmC binding are shown in the enlarged part, and the 5hmC recognition of the SRA domain of UHRF2 (PDB ID 4PW6) is shown for comparison. **b** Structural superposition reveals a conserved ligand-binding conformation. **c** Measured distances between 5hmC-binding sites across CMoRE tetramers. The CMoRE tetramers are shown in surface and the 5hmC molecules are shown in balls. **d** Proposed model for the anti-phage mechanism of the CMoRE system.

mechanism for DNA degradation, we assessed its endonuclease activity against ordinary DNA in the presence of either 5hmC-modified DNA or 5hm-dCTP. Our results demonstrate that neither 5hmC-containing DNA nor 5hm-dCTP stimulates cleavage of ordinary DNA (Supplementary Fig. 8), arguing against an allosteric activation mechanism.

We then propose a mechanistic model for the defense mechanism of CMoRE against T-even phages (Fig. 5d). Upon the infection of T-even phages, the modified viral genomic DNA is recognized by the C-terminal domain of CMoRE and degraded by its N-terminal nuclease domain. The unique negatively-charged loop observed in CMoRE may act like a safety-catch to avoid unspecific degradation of the host DNA. Since the affinity of CMoRE for 5hmC- or 5ghmC-modified DNA is much higher than 5mC-modified or ordinary DNA (Fig. 2e), the 5hmC- or 5ghmC-modified phage genomic DNA will bind much more tightly to CMoRE and thus have enough time to outcompete the negatively-charged loop to approach the active site and get degraded.

### CMoRE represents an additional member of GIY-YIG family

To analyze the distribution of the CMoRE system, we used DefenseFinder[1] to identify bacterial and archaeal CMoRE systems against 22,803 prokaryotic genomes. A total of 274 CMoRE systems have been identified, which are phylogenetically widespread across 84 distinct genera spanning 15 classes (Fig. 6a, b). All CMoRE systems are single-component. Based on the domain architecture, the identified CMoRE systems could be clustered into four types (Fig. 6c). The two CMoRE systems reported here belong to type I. Type II, III and IV CMoRE systems contain an additional domain on either N-terminus or C-terminus. All the identified CMoRE systems have the third tyrosine residue to form a "GIYxY-YIG" motif and most have the negatively-charged loop (Fig. 6d), and thus CMoRE represents an additional member of GIY-YIG family.

## Discussion

Arose out of the evolutionary arms race between bacteria and bacteriophages, the RM systems protect bacteria from phage infection. As a countermeasure, DNA modifications, such as 5hmC and 5ghmC, render viral DNA resistant to cleavage by the endonucleases of RM systems. Here, we reported a bacterial defense system, CMoRE, which specifically recognized and degraded 5hmC- or 5ghmC-modified DNA

to mediate antiviral immunity, highlighting the evolutionary interplay between bacteria and phages.

This defense system has been shown to protect against phages of the Tevenvirinae subfamily[15] and recognize modified phage DNA to elicit phage defense[16]. We combined biochemical and structural approaches to demonstrate that CMoRE is a type IV MDRE: sensing DNA modifications by its C-terminal domain and degrading target DNA by its N-terminal GIY-YIG nuclease domain.

The GIY-YIG domain has been found in all kingdoms of life[17]. We show that the GIY-YIG domain of CMoRE represents an additional member of GIY-YIG family since it displays two unique features. First, it has a third tyrosine residue between the two conventional tyrosine residues of the "GIY-YIG" motif, forming a "GIYxY-YIG" motif (Fig. 6d). Second, there is an inserted negatively-charged loop covering the active site of CMoRE's GIY-YIG domain, likely playing an autoinhibitory role to avoid autoimmunity and contribute to substrate specificity (Fig. 6d).

No homologues have been found for CMoRE's C-terminal modification-sensing domain by sequence or structure search, indicating a distinct protein fold of 5hmC/5ghmC recognition. The 5hmC molecule binds to the edge rather than a pocket of the modification-sensing domain (Fig. 5a). This binding mode is compatible with CMoRE's recognition for both 5hmC and 5ghmC, since a pocket is hard to accommodate both the small 5hmC and the much larger 5ghmC with an attached glucose moiety (Supplementary Fig. 9). It has been suggested that CMoRE also recognizes carbamoylmethyl-modified adenines[16]. Recently, 5-arabinose-hydroxy-cytosine modification was identified in some phages[18]. We therefore investigated whether CMoRE could resist Bas46 and Bas47 phages, which feature double arabino-sylation of hydroxy-cytosines[18]. Plaque formation assays revealed that CMoRE exhibited only very mild resistance to these phages (Supplementary Fig. 10a). Consistent with this, CMoRE displayed significantly reduced endonuclease activity against the genomic DNA of Bas46 or Bas47 compared to 5ghmC-modified T4 genomic DNA (Supplementary Fig. 10b). Together, these findings suggest that CMoRE exhibits substrate recognition plasticity, cleaving a spectrum of cytosine modifications with differential efficiencies.

Crystal structures revealed a tetrameric assembly for CMoRE (Fig. 4). We have observed a same tetrameric configuration for CMoRE in two species (Fig. 4a), implying such tetramer formation is not artificial. Dimeric or tetrameric architectures are often found in restriction

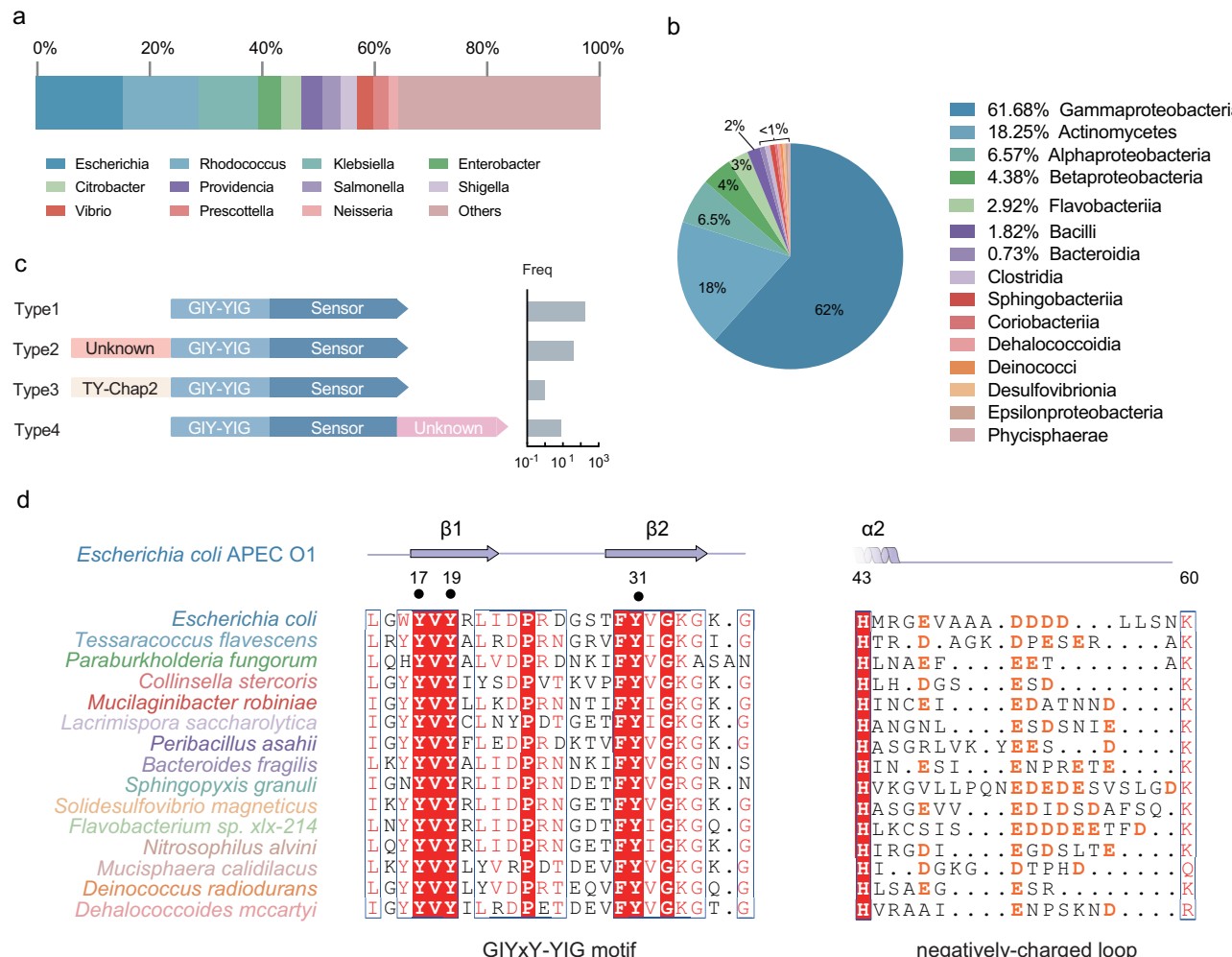

**Fig. 6 | CMoRE represents an additional member of GIY-YIG family.**
**a** Phylogenetic distribution of bacterial genera encoding CMoRE systems.
**b** Taxonomic class-level distribution of bacteria encoding CMoRE systems.
**c** Domain architectures and classification of the CMoRE family members. Predicted domains are shown using different colors and shapes. The number of systems belonging to each type is indicated. **d** Multiple sequence alignment of representative CMoRE proteins across the 15 classes. Only the segments of the GIYxY-YIG motif and the negatively-charged loop are shown. The secondary structure elements of *E. coli* APEC O1 CMoRE are shown above the alignment. Source data are provided with this paper.

endonucleases, since they usually interact with a symmetric recognition site and make a double-strand break[19–21]. For CMoRE, it seems that a CMoRE dimer is required for the symmetric recognition of two 5hmC sites (Figs. 2g and 5c). The tetrameric assembly presents two CMoRE dimers providing potential recognition for two DNA substrate molecules.

DNA methylation occurs at the C-5 position of cytosine in most eukaryotic organisms, resulting in 5-methylcytosine (5mC)[22,23]. The oxidation product of 5mC, 5hmC, has been proposed as a promising epigenetic marker for the diagnosis, treatment, and prognosis of various diseases, including cancers, neurological disorders, as well as autoimmune diseases[5]. However, the detection of 5hmC at the genomic level poses a significant challenge due to its low abundance[24]. Great efforts have been made for the precision quantification of 5hmC, and high-resolution, high-sensitivity and cost-effective methods based on restriction endonucleases (REs) have been developed to achieve genome-wide 5hmC mapping at single-base resolution[25]. Nevertheless, low substrate specificity limits this application. A major technical challenge is that 5hmC is structurally and chemically very similar to 5mC (Supplementary Fig. 9). For example, McrBC and MspJI could recognize both 5mC and 5hmC[26,27]. PvuRts1I specifically recognizes 5hmC as well as 5ghmC, and has been used as a tool to map 5hmC in mammalian genomes[28–30], since 5ghmC does not exist in eukaryote.

However, purified PvuRts1I protein is unstable, sensitive to high concentrations of imidazole, Cl⁻, or prolonged incubation at room temperature[28,29]. Here we show that CMoRE is a stable restriction endonuclease, which specifically recognizes 5hmC and 5ghmC, but not 5mC or ordinary cytosine (Fig. 2), providing a potential tool for 5hmC detection in mammals.

## Methods

### Plasmid construction
The genes of CMoRE systems derived from *Escherichia coli* O157:H7 (WP_000355475.1), *Escherichia coli* APEC O1 (WP_000353910.1), and *Klebsiella pneumoniae* 30684/NJST258_2 (AHM78501.1) were synthesized and cloned into the pET28a vector containing an N-terminal 6×His tag followed by a tobacco etch virus (TEV) protease recognition site. All point mutations were prepared by PCR-based site-directed mutagenesis using the wild-type constructs as the templates.

### Phage propagation and efficiency of plaquing assays
Bacteriophage stocks were purchased from Deutsche Sammlung von Mikroorganismen und Zellkulturen (DSMZ): T1 (DSM 5801), T2 (DSM 16352), T3 (DSM 4621), T4 (DSM 4505), T5 (DSM 16353), T6 (DSM 4622), T7 (DSM 4623), and P1(DSM 5757). For phage propagation, a single phage plaque was picked and inoculated into a liquid culture of

*E. coli* BL21(DE3) of mid-log phase and incubated at 37 °C with shaking until complete culture lysis. Lysates were clarified by centrifugation at 3040 × *g* for 10 min, followed by sterile filtration using a 0.22-μm filter. Phage titer was determined by the small drop plaque assay.

Overnight cultures of *E. coli* cells harboring the defense system or empty vector were diluted 1:50 in LB medium and incubated at 37 °C with shaking until early log phase (OD$_{600}$ = 0.3). For the phage plaque assay, 2 mL culture was mixed with 23 mL melted LB agar (LB + 0.6 % agar w/v, supplemented with 50 μg mL$^{-1}$ kanamycin) and poured into a square plate. The plates were solidified at room temperature for 1 h. The phage stocks were subjected to tenfold serial dilutions in LB medium and 3-μL drops of each dilution were spotted onto the bacterial lawn. After droplets absorption, the plates were inverted and incubated overnight at 37 °C. Plaque-forming units (PFUs) were quantified by counting the plaques following overnight incubation.

### Liquid culture growth assays
Early-log culture (180 μL per well) was dispensed into a 96-well plate containing 20 μL phage suspension for a final MOI of 2 or 0.02, with LB broth serving as an uninfected control. Biological triplicates were established using three independent overnight cultures derived from distinct single colonies. Plates were incubated at 37 °C with continuous shaking in SpectraMax® Absorbance Plus Microplate Readers (Molecular Devices) and OD$_{600}$ values were measured at 5-min intervals over 8 h.

To detect the cell toxicity, early-log cultures was induced by IPTG and dispensed into 96-well plates (200 μL per well) and incubated at 37 °C with shaking. Bacterial growth was monitored by measuring OD$_{600}$ at 5 min intervals over 6 h using SpectraMax® Absorbance Plus Microplate Readers (Molecular Devices).

### Protein expression and purification
*E. coli* BL21(DE3) cells harboring the expression plasmid were cultured in LB supplemented with kanamycin (50 μg mL$^{-1}$) at 37 °C with shaking. A single transformant colony was cultured in 20 mL LB overnight. Then, the culture was inoculated into 1 L fresh LB medium for large-scale cultivation. Protein expression was induced at mid-log phase (OD$_{600}$ 0.6–0.8) with 0.2 mM isopropyl β-D-1-thiogalactopyranoside (IPTG), followed by incubation at 18 °C for 16–18 h. Cells were harvested by centrifugation at 3040 × *g* for 20 min at 4 °C, resuspended in lysis buffer (20 mM Tris-HCl pH 8.0, and 500 mM NaCl) and lysed using an ultrahigh-pressure homogenizer. Supernatant was collected after centrifugation, followed by filtration with a 0.45 μm membrane prior to loading onto a Ni-NTA column that had been pre-equilibrated with lysis buffer. Sequential washes were performed with lysis buffer containing 10 mM and 20 mM imidazole, followed by elution with lysis buffer containing 250 mM imidazole. The N-terminal tag was removed by TEV protease at 4 °C overnight. The digested protein was passed through a HisTrap™ HP column (Cytiva) to remove free tag and TEV protease. The flow-through was collected and further purified by size-exclusion chromatography using Superdex™ 75 10/300 GL column (Cytiva) equilibrated with buffer containing 20 mM HEPES pH 7.5, 200 mM NaCl and 1 mM DTT. The protein purity was verified by sodium dodecyl sulfate-polyacrylamide gel electrophoresis (SDS-PAGE) followed by Coomassie Brilliant Blue R-250 (Sangon Biotech) staining.

### Nucleic acid substrate preparation
Genomic DNA from bacteriophage and bacteria were extracted using the Lambda phage Genomic DNA Kit (Zoman Biotech) and TIANamp Bacteria DNA Kit (TIANGEN), respectively. To prepare DNA substrates for restriction activity assays, three differentially modified fragments were PCR-amplified from T4 phage genomic DNA using dATP/dGTP/dTTP supplemented with dCTP, 5m-dCTP, or 5hm-dCTP, respectively (1,026-bp, C/modified-C: 0.58%/17.60%). The 5ghmC-modified substrate was enzymatically synthesized from 5hmC-DNA using T4

β-glucosyltransferase (NEB) according to manufacturer specifications. All amplified products were verified by agarose gel electrophoresis and purified by the Universal DNA Purification Kit (TIANGEN).

### Nuclease activity assays
Nuclease activity assays were conducted in a reaction volume of 10 μL containing 200 ng substrate DNA and 1 μM CMoRE proteins in rCutSmart™ Buffer (New England Biolabs). Enzymatic reactions were carried out at 37 °C for 20 min and terminated by DNA loading dye. Reaction products were resolved by agarose gel electrophoresis and visualized using a Gel Doc™ XR+ system (Bio-Rad).

### Quantitative PCR
Overnight cultures of *E. coli* cells were diluted 1:50 using fresh LB medium. The diluted cultures were grown at 37 °C to mid-log phase OD$_{600}$ 0.6–0.8 and subsequently infected with T4 phage at an MOI of 4. Samples were collected at 2 min, 4 min, and 8 min post infection by centrifugation at 21,270 × *g* for 2 min at 4 °C. Supernatants were removed and pellets immediately frozen at −80 °C for DNA extraction later.

Total DNA was extracted using the TIANamp Bacteria DNA Kit (TIANGEN). Phage T4 DNA was quantified by real-time PCR using AceQ Universal SYBR qPCR Master Mix (Vazyme) on a CFX Connect Real-Time PCR Detection System (Bio-Rad), with T4-specific primers (F: ATCCATCGTGATCTGCGTCT, R: TAAACGCGGTTGGATTCCTG) and 16S rRNA endogenous control primers following the manufacturer's protocol. For data analysis, the relative phage DNA abundance was determined using the ΔCt method, with values normalized to the endogenous 16S rRNA control.

### DNA library preparation and DNA sequencing
Genomic DNA extracted from T4 phage-infected samples (8 min post infection) were processed using the Illumina TruSeg® Nano DNA High Throughput Library Prep Kit according to manufacturer's instructions. For cleavage site mapping, 1 μg T4 genomic DNA was incubated with 1 μM CMoRE protein in 1× rCutSmart Buffer (New England Biolabs) at 37 °C for 5 min, followed by ethanol purification. Digested fragments were converted to sequencing libraries using the NEBNext Ultra II DNA Library Prep Kit for Illumina. Library quality was assessed by Qubit fluorometric quantification (Thermo Fisher Scientific) and fragment size distribution analysis (Agilent 4200 TapeStation). Libraries were sequenced (2 × 150 bp paired-end) on an Illumina MiSeq platform.

Raw sequencing data underwent a standardized bioinformatics pipeline: (1) quality control with fastp (v0.25.0) including adapter trimming and quality filtering (Q20 threshold); (2) alignment to the T4 reference genome (NC_000866.4) using BWA-MEM (v0.7.18); (3) duplicate read removal using SAMtools (v1.21); and (4) cleavage site identification through precise mapping of fragment ends. For cleavage site analysis, 20 bp flanking sequences from aligned read termini were extracted using bioawk (v1.0) and mapped to nucleotide positions with single-base resolution.

### Electrophoretic mobility shift assay
PCR-amplified DNA fragments (100-bp, C/modified-C: 5.5%/14.5%, 0.1 μM) were incubated with CMoRE protein at concentrations of 0, 0.1, 0.2, 0.5, 1, 2, and 5 μM in a buffer containing 20 mM Tris-HCl (pH 8.0), 150 mM NaCl, 1 mM DTT and 5 mM EDTA for 30 min at 37 °C. The binding reactions were resolved on a 6% native polyacrylamide gel using 0.5×TBE running buffer. Gels were stained with GelRed nucleic acid dye for 1 min at room temperature and imaged using a ChemiDoc™ MP Imaging System (Bio-Rad).

### Protein crystallization, X-ray data collection and structure determination
Purified protein was concentrated to 10 mg mL$^{-1}$ in 20 mM HEPES pH 7.5, 200 mM NaCl, and 1 mM DTT. Crystallization trials were performed

at 20 °C using the hanging-drop vapor-diffusion method. The crystals of *E. coli* O157:H7 CMoRE and CMoRE/5hm-dCTP complex were grown in a buffer consisting of 0.1 M MES pH 6.0, 14% w/v Polyethylene glycol 4000. The crystals of *E. coli* APEC O1 CMoRE were grown in a buffer consisting of 0.49 M Sodium phosphate monobasic monohydrate, 0.91 M Potassium phosphate dibasic, pH 6.9. The crystals of *E. coli* APEC O1 CMoRE/5hm-dCTP complex were grown in a buffer consisting of 0.1 M HEPES pH 7.5, 800 mM Potassium phosphate dibasic, 800 mM Sodium phosphate monobasic. Crystals were cryoprotected in 40% sucrose and flash-frozen in liquid nitrogen. X-ray diffraction data were collected at the beamline BL19U1 at the Shanghai Synchrotron Radiation Facility (SSRF). Data were processed using HKL3000[31] or XDS[32]. The initial models of CMoRE were predicted by AlphaFold[33]. Structure refinement and model building were performed with PHENIX[34] and Coot[35]. All the models were validated using MolProbity[36]. Details of the refinement statistics were summarized in Supplementary Table 1. Images of representative electron density for each structure were provided in Supplementary Fig. 11. All the structural figures were generated using ChimeraX-1.5[37] and PyMOL (https://pymol.org/2/).

### Bacterial toxicity assay
Toxicity assays were performed using the IPTG-inducible pET28a vector in *E. coli* BL21(DE3). Overnight cultures of *E. coli* BL21(DE3) harboring the constructs were grown in LB medium with 50 µg mL$^{-1}$ kanamycin, diluted 1:50 in fresh LB, and incubated at 37 °C with shaking for 1.5 h. Cultures were serially diluted tenfold with LB medium and spotted onto LB agar plates (0.6% agar) containing 50 µg mL$^{-1}$ kanamycin and 0.02 mM IPTG. Plates were incubated at 37 °C for 16 h before imaging.

### Statistics and reproducibility
All experiments were independently repeated three times with similar results.

### Reporting summary
Further information on research design is available in the Nature Portfolio Reporting Summary linked to this article.

## Data availability
Atomic coordinates and structure factors in this study have been deposited in the Protein Data Bank (PDB) under accession codes: 9U7U (*E. coli* APEC O1 CMoRE), 9U7Z (*E. coli* APEC O1 CMoRE/5hm-dCTP), 9U75 (*E. coli* O157:H7 CMoRE), and 9U8D (*E. coli* O157:H7 CMoRE/5hm-dCTP). The sequencing raw data generated in this study have been deposited to Genome Sequence Archive (GSA) under accession code CRA027481 (T4 phage genome in *Escherichia coli* without PD-T4-3), CRA027480 (T4 phage genome in *Escherichia coli* with PD-T4-3), CRA027479 (T4 phage genome degraded by PD-T4-3). Source data are provided with this paper.

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

## Acknowledgements
We thank the staffs from BL19U1 beamline of National Center for Protein Science Shanghai (NCPSS) at Shanghai Synchrotron Radiation Facility for assistance during data collection. We thank Dr. Pan Tao for providing the genetically engineered T4 phages. We thank Dr. Alexander Harms and Dr. Wenyuan Han for providing Bas46 and Bas47 phages. Financial support for this work was provided by National Key Research and Development Program of China (2025YFC3408400 to Y.Y. and 2024YFE0106200 to Z.Z.), National Natural Science Foundation of China (32270761 to Q.C. and 32400125 to D.T.), Sichuan Science and Technology Program (2024NSFTD0029 to Q.C. and 25QNJJ5253 to R.L.), and Sichuan Foundation for Distinguished Young Scholars (25NSFJQ0249 to Y.Y.).

## Author contributions
Q.C., Y.Y. and Z.Z. conceived and designed the experiments. R.L., D.T., and M.N. performed experiments. S.L. analyzed the sequencing data. Q.C. and Y.Y. wrote the manuscript.

## Competing interests
The authors declare no competing interests.
