## [Transparent Peer Review file · Nature Communications]

A bacterial defense system targeting modified cytosine of phage genomic DNA

Corresponding Author: Dr Qiang Chen

Version 0:

Reviewer comments:

Reviewer #1

(Remarks to the Author)

The discoveries of anti-phage defence systems have rapidly increased in the past years, and the mechanisms of most recently discovered systems remain elusive. The manuscript from Lui, Tang et al. investigates the mechanism of the PD-T4-3, an anti-phage defence system from *Escherichia coli* reported by Vassallo et al. in 2022 (Nature Microbiology), which mechanism of action remains unknown. Using approaches from structural biochemistry, the authors report that PD-T4-3 is an endonuclease that cleaves modified phage DNA, specifically hydroxymethyl cytosine (hmC), or glucosylated hmC, a modification that is commonly found in T-even phages. Moreover, the authors report the structure of PD-T4-3, giving further insight on its molecular mechanism of action.

The work presented in the manuscript is generally sound and will be of interest for the scientific community interested in prokaryotic immunity.

My main criticism concerns the somewhat limited depth of the investigation into PD-T4-3 activity, which, considering recent publications, may be expanded to enhance the generalizability of the findings.

While PD-T4-3 has been uncovered recently, there are at least two studies (preprints) that investigate its activity, but none of these papers have been mentioned in the manuscript (doi: 10.1101/2025.02.28.64065 ; doi : 10.1101/2024.12.20.629700). In particular, one study provides direct evidence of the activity of PD-T4-3 against modified phage DNA and further suggests that PD-T4-3 also has some activity on phage DNA with methyl-N6A modification (doi: 10.1101/2025.02.28.64065).

Interestingly, a recent paper has reported that phages related to phage T4 may have 5-arabinose-hydroxy-C modifications (<https://doi.org/10.1016/j.chom.2025.06.005>), with some phage families carrying single, double, or triple arabinosylated cytosines. It would be interesting to know whether PD-T4-3 cleaves DNA with such modifications.

Given that the activity of PD-T4-3 may be broader and more plastic than expected, investigating the breadth and specificity of PD-T4-3 activity in more details could strengthen this study.

Minor points are listed below

Line68-70: What is the level of divergence between these three genes? What is the rationale for choosing these 3 genes? Can the weaker activity of PD-T4-3 from APEC O1 be linked to sequence divergence in the coding sequence or in the promoter?

It should be explicitly mentioned that the genes are cloned on a plasmid and therefore, are overexpressed. The expression level of defence systems is critical for their activity.

Line73-74: Please note that "Caudovirales, Myoviridae, Podoviridae" are no longer used for phage classification

Figure S1

Note that "Efficiency of Plating" refers to a ratio (e.g. phage PFU on strain with defence/phage PFU on strain with empty vector). Figure S1 does not show EOP (y-axis labelling)

Figure 2b

Specify which phage gene is measured in qPCR

Because T4 depletes host DNA, normalisation against bacterial housekeeping gene is biasing the interpretation of variation in the relative amount of phage DNA during infection; i.e., the amount of host DNA is likely low in the qPCR made on infected cells with empty vector while it is higher in the condition with the vector encoding PD-T4-3. A control experiment

where similar qPCR are done in cells infected by phages that are not targeted by PD-T4-3 (e.g. T5) would limit the bias in the interpretation of the results,

Figure 2c

To fully interpret the data, it would be interesting to see the plot of DNA reads matching bacterial genome

If the 4A mutant of PD-T4-3 efficiently degrades genomic DNA, it is surprising that the toxicity is not higher than a 2-log reduction in cell viability. I think this should be commented.

Line 189. Define SRA domain and UHRF2

Line 216 – 274. It is mentioned that 274 PD-T4-3 systems have been identified. How many genomes were analysed? This should be specified to provide a better idea of the frequency of this system in known genomes.

Reviewer #2

(Remarks to the Author)

The Manuscript by Liu et. al. describes PD-T4-3, an unusual and novel family of GIY-YIG endonucleases that specifically recognise modified nucleotides present in phage DNA. The authors have characterised three members of this family, first showing that expression of the endonuclease conveys resistance of the host to T-even phages, and then following up with a thorough molecular characterisation and structural determination of members of the family.

The novelty of the presented work is in the unusual modification sensing domain of the GIY-YIG nuclease family, which is able to stringently distinguish between cytosine and 5hmC/5ghmC. By determination of structures bound to modified nucleotide, the authors are able to postulate how the sensing occurs, and how this primes the nuclease for cleavage of DNA containing these modifications. However, the authors have not presented a convincing account of how the sensing domain of PD-T4-3.

The biochemical data is robust and well presented, showing the preference of PD-T4-3 for modified cytosine over unmodified cytosine, and also testing metal ion preference and a number of key mutations to active site and the autoinhibition loop. The authors also show convincingly the effect of introducing this system into E. coli lacking the system, and that not only does it protect from T-even phage infection, but also depletes phage DNA within infected cells. The authors also show that lacking the autoinhibitory loop, the expression of 4A mutant is detrimental to E. coli, showing this loop is important for control of the nuclease. However, the authors find no correlation between binding of modified cytosine in the sensing domain and relief from the autoinhibitory loop in order to activate DNA hydrolysis.

The structures of PD-T4-3, and especially from two related systems, helps to convince on the biological relevance of the dimeric and tetrameric assembly, and the binding mode of the 5hmC present in both co-crystal structures. However, some of the structural data, and specifically the ligand bound structure do not convince me about the mechanism of action of the system determined with the biochemical data. Two major issues are present, and should be expanded upon. The first issue is the relatively weak and incomplete density of the ligand. The authors should present scoring from validation software or report metrics (such as bonding network and/or B-factors of ligand atoms and surrounding bound protein atoms) to allow the reader to decide on the density quality. Softwares such as 'Checkmyblob' would also help to verify the ligand density is interpreted correctly. The second issue is that the crystal structure contains only a single nucleotide rather than DNA incorporating the modified nucleotide. The issue with this is that the available surface in a DNA duplex containing the modified nucleotide is vastly different from that present in the single nucleotide. The authors suggest a similar binding mode in UHRF2-DNA structure, but more comparison and description is required to convince a reader this is the case.

The authors should consider the following corrections in order to strengthen the manuscript, and to present more clearly their findings.

Major Comments

The authors do not report the methods used to co-crystallise PD-T4-3 with 5hmC. Given the density presented in the paper (Fig 5a), it would be critical to know the details of crystallisation for these structures. More comparison between the apo and co-crystal structure would also be insightful

In the validation reports, the density for the O157:H7 PD-T4-3 5hmC looks to be of a better overall quality, covering more of the ligand. Could the authors show this density also in Fig 5a. The resolution of APEC O1 is higher than that of O157:H7, but the density for the ligand is much worse. Could the authors comment on this?

Figures containing structures are difficult to interpret (e.g. fig4, fig5). Insets are busy and the slabbing such that much of the insets are almost transparent.

Figure 4c insets - difficult to see details of insets due to small size and clipping such that many details are too transparent.

Authors should enlarge to show more clearly the details.

Figure 5a inset - show a second inset with the structures overlaid in order to show that the arrangement of the nucleotides are similar in each structure.

A figure showing the autoinhibitory loop position in the modified cytosine bound structure would be insightful, as well as a comparison of the apo and ligand bound structure. If no conformational change is seen, a description to this effect would be insightful. Is there any relative motion of units within the tetramer upon binding the ligand?

Minor comments

The description of the additional tyrosine on In140 is confusing. The authors explain the situation better in the discussion, and so the first reporting could be reworded for clarity as in the discussion.

Fig 6d - show the lack of secondary structure above sequence alignment as a line rather than gaps.

Reviewer #3

(Remarks to the Author)

This paper from Liu et al examines a recently identified anti-phage defense system provisionally called PD-T4-3. The authors here show that the C-terminal domain of PD-T4-3 specifically recognizes phage DNA carrying hydroxymethylated cytosines with the N-terminal GIY-YIG domain then cleaving the DNA, making PD-T4-3 a variant of type IV restriction enzymes. Although the paper provides new insights into this anti-phage defense system, significant questions remain about the structural and mechanistic basis by which the system discriminates phage DNA from host DNA. These questions and other issues are articulated below:

Major questions/issues:

- 1) The data presented here suggest that PD-T4-3 defends against T2, T4, and T6 equally well. But in the original paper that identified the system, defense is far stronger against T4 than against T2 and T6. This discrepancy warrants some explanation. Is it because the PD-T4-3 homologs here are different or because the host strain being used is different? Using the original PD-T4-3 system as a control would help address this question.
- 2) Is the recognition of both 5hmC and 5ghmC necessary for PD-T4-3 based defense? The gel shown in Fig 2d shows that both 5hmC- and 5ghmC-DNA are recognized, but what is the significance of this in vivo? It would be more convincing that these modifications are required for defense if the authors could show that deglycosylated strains of T-even phages are able to plaque better on PD-T4-3+ lawns. Additionally, is it possible to generate T-evens without 5hmCs as well (and then use a strain of E. coli without RM systems for plaquing)? If unmodified T-even phages are able to escape PD-T4-3, it would be more convincing that the modifications are necessary for PDT43 based defense.
- 3) Related to the above point, it's a little unclear how the C-terminal domain is specifically sensing 5hmCs or 5ghmCs. In the structure shown in Fig 5a, the authors show that the main chain oxygen of Y127 interacts with the hydroxyl group of the modified cytosine. They then later make a point that since the modified cytosine is bound to the edge of the protein and not a binding pocket, the C-terminal domain can sense both 5hmC and 5ghmC (lines 249-251). Could PDT46 potentially bind promiscuously to other hydroxylated base modifications found in phage DNA, such as arabinosyl-hmC from RB69?
- 4) How PD-T4-3 specifically targets phage DNA while sparing host DNA is still unclear. PD-T4-3 features a negatively charged loop that occludes the active site, leading the authors to speculate that it may serve to prevent degradation of the host chromosome. The authors propose a model in which binding of the C-terminal domain to modified DNA is then followed by phage DNA 'outcompeting' the loop to enable degradation. But it could be that binding to modified DNA leads to an active movement of the loop to enable cutting of phage DNA. A structural analysis of PD-T4-3 bound to a longer piece of DNA containing modified cytosines is probably needed to address this question and provide insight into how PD-T4-3 exclusively targets phage DNA. As it stands, the authors have only examined PD-T4-3 bound to 5hm-dCTP, which offers limited insight. The authors also might be able to test if a non-cleavable 5hmC DNA analog can prime cleavage of free DNA, which would support the allosteric model.
- 5) The authors state that the GIY-YIG nuclease domain of PD-T4-3 and its homologs is novel because it has an extra tyrosine, but it's not clear why this tyrosine is significant or what role it plays. A mutation in that tyrosine does disrupt defense, but the function/role is unclear.
- 6) The crystal structures reveal potential tetramerization, but the authors do not probe whether the interface formed between dimers or within a dimer are necessary for activity in vivo. Can mutations be designed based on the structure to test each interface and its relevance to defense? Also, can the oligomerization state of the protein be assessed by size exclusion chromatography to determine whether it forms a dimer, tetramer, both, or something else?

Minor comments:

- 1) The authors should rename PD-T4-3. Those names were provisional - follow up studies on other systems identified in that original screen have since been renamed so PD-T4-3 should probably also get a better name that ideally reflects its function/activity. DefenseFinder should be updated accordingly.
- 2) Line 54: it would be good to mention which E. coli strain PD-T4-3 was initially found in (ECOR68).
- 3) Lines 80-81: should indicate that these data confirm those already presented in Vassallo et al where defense as a function of MOI was reported for PD-T4-3.
- 4) Line 97: it's mentioned that the phage DNA was severely depleted across the entire genome. This also supports the idea that this nuclease has limited sequence specificity, which would be consistent with the next few panels in Fig 2. Maybe they can mention this at this stage?
- 5) Line 98, 99: it's unclear how the divalent cation preference is relevant here. Maybe it makes more sense to talk about metal dependency prior to mentioning any in vitro assays, as the cation preference is presumably incorporated into those assays?
- 6) Line 113-123: the methodology described here was a bit opaque. It would be helpful to include a small diagram explaining the process. Additionally, in the corresponding Fig. 2f the N9 and N11 sites appear to be incorrectly positioned in the cartoon representation of the DNA.
- 7) Line 107-111: Can the EMSA assays be used to calculate KD? And if the authors performed an EMSA for 5hmC DNA as well, maybe they can compare any differences in KD between these two substrates. In general, having KDs would help to reveal the strength of the interactions reported. Also, why doesn't the DNA get cleaved in the EMSAs?
- 8) Line 145: The finding that mutations in the catalytic residues reduce defense is compelling. It could be a nice correlation to show the same mutants are no longer capable of DNA cleavage using the previously described in vitro nuclease assay.
- 9) Line 149: What's the length and sequence of the autoinhibitory loop? Would help to include it here.
- 10) Line 160: Is the bacterial toxicity studied using an inducible construct? There didn't seem to be a methods section for how bacterial spotting was done.
- 11) Line 160-162: The text suggests that this nuclease may adopt an inactive and active conformation. It would be helpful if the authors could elaborate on this point more explicitly and maybe speculate about the differences in the two conformations.
- 12) Line 189: Perhaps could explain what UHFR2 and an SRA domain is—was unsure of the significance.
- 13) Line 220, 221: how do these other PD-T4-3 systems with additional domains differ? Do they have altered specificity or other activities?
- 14) Line 278: This last claim about PD-T4-3 being a 'reliable tool for 5hmC detection in mammals' is an overstatement as there are no data about how PD-T4-3 interacts with eukaryotic DNA.
- 15) Many sentences could be rewritten to improve clarity and there were a variety of grammatical issues throughout.

Comments on Figures:

Fig 1a: Should include the length of the gene. Or maybe protein product size/individual domain sizes?

Fig 2b: indicate what T4 gene is used for qPCR analysis (did not appear to be included in the methods).

Fig 2c: is missing an x-axis label.

Fig 2d, 2e: indicate what PCR product is being used for the nuclease activity assays and EMSAs. In the methods section the authors mention these fragments are amplified from the T4 genome, but it's unclear what the length of these fragments is and their cytosine/modified cytosine content. Same issue/question for Fig 3g.

Fig 3b. legend: "α2-α3 loops are highlighted by deep colors" should probably say "dark colors".

Version 1:

Reviewer comments:

Reviewer #1

(Remarks to the Author)

The authors have improved their manuscript upon revision.

While I maintain that evaluating the breadth of CMoRE activity would certainly strengthen the paper - a view that is shared by Reviewer 3 - the authors explain that this exploration is not possible due to technical challenges or unavailability of biological material. Although this is a valid reason, I believe that phages Bas46 and Bas47 could be obtained from other labs - with Dr. HARMS permission - as the BASEL collection is now so widely distributed.

Regarding my comment about normalisation bias in qPCR assays:

I apologize if my previous comment was unclear. I was not suggesting to co-infect cells with T4 and T5 for the qPCR assay, which may introduce other problems. I was rather suggesting to perform qPCR assay on cells infected by T5 (not targeted by CMoRE) to monitor T5 DNA replication. The expectation with T5 is that phage DNA will increase over time, even in the presence of CMoRE. This is to show that variations in relative amount of phage DNA is not driven by host DNA depletion. Perhaps a better control would be to use a phage targeted by CMoRE but that is not inducing host DNA depletion but it is probably difficult to find a phage that matches these criteria.

That being said, looking now at the new panel in Fig 2c, the quantity of host DNA 8min post-infection is comparable in the two strain backgrounds which probably limits normalization bias I was expecting. In the light of this data, I would suggest that the authors restore their initial Fig2b.

My other comments have been satisfactorily addressed.

Reviewer #2

(Remarks to the Author)

The authors have made the necessary changes to the manuscript, and I am happy with the changes made.

Reviewer #3

(Remarks to the Author)

The responses and additional experiments from the authors have improved the clarity and strength of the paper. In particular, the authors now provide better support for the conclusion that T4 containing unmodified cytosines (deglycosylated T4(hmC)) is completely able to evade CMoRE. The *in vivo* experiments suggest that recognition of 5hmC alone is sufficient to trigger complete defense, which suggests that the variability in defense against different T-even phages seen in some CMoRE homologues is likely attributable to factors other than glycosylation — perhaps differences in genome architecture or in the time scale of infection. The authors also now provide some additional data on a priming experiment to demonstrate that CMoRE does not employ an allosteric mechanism for DNA degradation. This helps explain better how CMoRE avoids autoimmunity. There are, however, some remaining issues with the manuscript:

1) As originally noted, the PD-T4-3/CMoRE system used here differs in its defense profile compared to that first published. In their responses the authors argue that this is because they are using a different homolog. I'm confused - is that really the case? Maybe I missed it, but it seemed initially like it was the same homolog/protein. If that's not the case, the authors should provide an alignment (unless I missed it) and describe the relationship of their homolog to the original one.

2) I still think the authors should test whether CMoRE can recognize arabinosylated cytosine. They indicate that they requested three such phages from Harms but don't yet have them. There are many other labs that have these phages and can likely provide them. Especially RB69, which isn't even part of the BASEL/Harms collection.

3) The role of the additional tyrosine isn't elucidated. The authors argue it's 'beyond the scope of the current study' but this is noted even in the Abstract as one of the key features of this study ("Unique features, including a 'GIYxY-YIG' motif and an inhibitory negatively charged loop, distinguish CMoRE28 as a novel member of the GIY-YIG family.")

4) It's interesting that some CMoRE catalytic mutants—despite completely abolishing endonuclease activity *in vitro*—did not seem to result in a very significant loss of defense *in vivo* (as shown in Fig. 3d and 3e). What is the explanation for this apparent discrepancy?

Minor comments:

In line 81, maybe the mechanism could be described as a "direct defense mechanism" rather than "phage-targeting".

In the section describing the mutations used to disrupt oligomerization (lines 190–196), it might be helpful to clarify that the R180A mutant did not alter oligomerization (as shown in Fig. 4f), which is consistent with the lack of impact on anti-phage activity.

Version 2:

Reviewer comments:

Reviewer #3

(Remarks to the Author)

The authors have made some additional changes and updates that improve the manuscript, which I support publication of at this stage.

REVIEWER COMMENTS

We highly appreciate the reviewers for their detailed and constructive comments that have significantly improved our manuscript. We have carefully revised our manuscript according to the reviewers' suggestions. In this revision, we have provided new data, including qPCR (Fig. 2b), DNA sequencing analysis (Fig. 2c), EMSA (Fig. 2e), *in vitro* activities of the mutants (Fig. 3e), mutagenesis of the tetramer interfaces (Fig. 4e & 4f), structural comparison and analysis (Fig. 5b & S7), genetically engineered T4 phages (Fig. S4), and evaluation of the allosteric mechanism (Fig. S8).

Reviewer #1 (Remarks to the Author):

The discoveries of anti-phage defence systems have rapidly increased in the past years, and the mechanisms of most recently discovered systems remain elusive. The manuscript from Lui, Tang et al. investigates the mechanism of the PD-T4-3, an anti-phage defence system from *Escherichia coli* reported by Vassallo et al. in 2022 (*Nature Microbiology*), which mechanism of action remains unknown. Using approaches from structural biochemistry, the authors report that PD-T4-3 is an endonuclease that cleaves modified phage DNA, specifically hydroxymethyl cytosine (hmC), or glucosylated hmC, a modification that is commonly found in T-even phages. Moreover, the authors report the structure of PD-T4-3, giving further insight on its molecular mechanism of action.

The work presented in the manuscript is generally sound and will be of interest for the scientific community interested in prokaryotic immunity.

We thank reviewer #1 for the positive comments.

My main criticism concerns the somewhat limited depth of the investigation into PD-T4-3 activity, which, considering recent publications, may be expanded to enhance the generalizability of the findings.

While PD-T4-3 has been uncovered recently, there are at least two studies (preprints) that investigate its activity, but none of these papers have been mentioned in the manuscript (doi: 10.1101/2025.02.28.64065 ; doi : 10.1101/2024.12.20.629700). In particular, one study provides direct evidence of the activity of PD-T4-3 against modified phage DNA and further suggests that PD-T4-3 also has some activity on phage DNA with methyl-N6A modification (doi: 10.1101/2025.02.28.64065). Interestingly, a recent paper

has reported that phages related to phage T4 may have 5-arabinose-hydroxy-C modifications (<https://doi.org/10.1016/j.chom.2025.06.005>), with some phage families carrying single, double, or triple arabinosylated cytosines. It would be interesting to know whether PD-T4-3 cleaves DNA with such modifications.

Given that the activity of PD-T4-3 may be broader and more plastic than expected, investigating the breadth and specificity of PD-T4-3 activity in more details could strengthen this study.

We thank reviewer #1 for pointing out this issue. In the revised manuscript, we have cited the two preprints (already published) and the 5-arabinose-hydroxy-C modifications paper (ref. 15, 16 and 18). We also added a paragraph to discuss this issue in Discussion section. Rodriguez-Rodriguez et al. indicated that PD-T4-3 recognizes T4 DNA glucose-modified cytosines and Mu DNA with carbamoylmethyl-modified adenines (ref. 16). However, they only use phage genomic DNA as the substrate which could not exclude the possibility that PD-T4-3 recognizes other DNA modifications instead glucose-modified cytosines or carbamoylmethyl-modified adenines. In our study, we use PCR products specifically containing 5hmC or 5ghmC modification (Fig. 2d), providing direct and more precise evidence for the substrate specificity of PD-T4-3. However, the commercial carbamoylmethyl-modified adenines or arabinosylated cytosines are not available.

We have sent e-mails asking for the phage strains with arabinosylated cytosine modifications (RB69, Bas46, and Bas47), but have not yet received a response. We notice that Bas46 and Bas47 are included in the BASEL collection (*PLoS Biology* 2021), therefore we contact Dr. Alexander Harms. Dr. Harms kindly replied to me saying that they need to generate new BASEL copies for shipping, but this may take 2-3 months since they have quite a backlog of earlier requests. The potential broader specificity for PD-T4-3 activity is really an interesting research question and we are eager to investigate it once the strains are accessible.

Minor points are listed below

Line68-70: What is the level of divergence between these three genes? What is the rationale for choosing these 3 genes?

The protein sequence identity between the three genes is 85%-88%. We perform the

anti-phage assays in *E. coli*, so we choose three species closely related to *E. coli*.

Can the weaker activity of PD-T4-3 from APEC O1 be linked to sequence divergence in the coding sequence or in the promoter?

The protein sequences of these three PD-T4-3 proteins are very similar (The protein sequence identity: APEC O1 vs O157:H7, 84.82%; APEC O1 vs *Klebsiella pneumoniae*, 88.72%; O157:H7 vs *Klebsiella pneumoniae*, 85.21%). Since replacing APEC O1's native promoter with T7 promoter resulted in a more robust phage resistance (Fig. 1b), it seems that the protein expression level of this defense system is linked to its anti-phage activity.

It should be explicitly mentioned that the genes are cloned on a plasmid and therefore, are overexpressed. The expression level of defence systems is critical for their activity.

We totally agree with reviewer #1 that the expression level is critical for the anti-phage activity of defense systems. In the revised manuscript, we have explicitly mentioned that the genes are cloned on a plasmid.

Line73-74: Please note that “Caudovirales, Myoviridae, Podoviridae” are no longer used for phage classification

We thank reviewer #1 for pointing out this issue. We have removed the description of the phage classification.

Figure S1

Note that “Efficiency of Plating” refers to a ratio (e.g. phage PFU on strain with defence/phage PFU on strain with empty vector). Figure S1 does not show EOP (y-axis labelling)

We thank reviewer #1 for pointing out this issue. We have corrected the Y-axis labelling in Fig. S1, as well as the Y-axis labelling in Fig. 3d.

Figure 2b

Specify which phage gene is measured in qPCR

Because T4 depletes host DNA, normalisation against bacterial housekeeping gene is biasing the interpretation of variation in the relative amount of phage DNA during infection; i.e., the amount of host DNA is likely low in the qPCR made on infected cells with empty vector while it is higher in the condition with the vector encoding PD-T4-3. A control experiment where similar qPCR are done in cells infected by phages that are not targeted by PD-T4-3 (e.g. T5) would limit the bias in the interpretation of the results,

For qPCR, we use a pair of primers specific to T4, not a specific gene. What really matters is the primer, not the amplified fragment. We have corrected the description in the figure legend and provided the primer sequence in the Methods part.

We really appreciate reviewer #1 for pointing out this issue of normalization bias. According to the reviewer's suggestion, we use both T4 and T5 at the same time for the qPCR assays, so that T5 depletes host DNA for both the system-lacking and the system-containing cells to limit the bias (Fig. 2b).

Figure 2c

To fully interpret the data, it would be interesting to see the plot of DNA reads matching bacterial genome

In the revised manuscript, we provide the plot of DNA reads mapped to the bacterial genome (Fig. 2c).

If the 4A mutant of PD-T4-3 efficiently degrades genomic DNA, it is surprising that the toxicity is not higher than a 2-log reduction in cell viability. I think this should be commented.

The 4A mutant of PD-T4-3 could degrade *E. coli* genomic DNA, but the efficiency is much lower compared to that of PD-T4-3 degrading T4 genomic DNA (Fig. S6). So, the toxicity of the 4A mutant is just moderate.

Line 189. Define SRA domain and UHRF2

We have defined SRA domain and UHRF2 in the revised manuscript.

Line216 - 274. It is mentioned that 274 PD-T4-3 systems have been identified. How many genomes were analysed? This should be specified to provide a better idea of the frequency of this system in known genomes.

We search PD-T4-3 against 22,803 prokaryotic genomes. We provide this information in the revised manuscript.

Reviewer #2 (Remarks to the Author):

The Manuscript by Liu et. al. describes PD-T4-3, an unusual and novel family of GIY-YIG endonucleases that specifically recognise modified nucleotides present in phage DNA. The authors have characterised three members of this family, first showing that expression of the endonuclease conveys resistance of the host to T-even phages, and then following up with a thorough molecular characterisation and structural determination of members of the family.

The novelty of the presented work is in the unusual modification sensing domain of the GIY-YIG nuclease family, which is able to stringently distinguish between cytosine and 5hmC/5ghmC. By determination of structures bound to modified nucleotide, the authors are able to postulate how the sensing occurs, and how this primes the nuclease for cleavage of DNA containing these modifications. However, the authors have not presented a convincing account of how the sensing domain of PD-T4-3.

The biochemical data is robust and well presented, showing the preference of PD-T4-3 for modified cytosine over unmodified cytosine, and also testing metal ion preference and a number of key mutations to active site and the autoinhibition loop. The authors also show convincingly the effect of introducing this system into E. coli lacking the system, and that not only does it protect from T-even phage infection, but also depletes phage DNA within infected cells. The authors also show that lacking the autoinhibitory loop, the expression of 4A mutant is detrimental to E. coli, showing this loop is important for control of the nuclease. However, the authors find no correlation between binding of modified cytosine in the sensing domain and relief from the autoinhibitory loop in order to activate

DNA hydrolysis.

The structures of PD-T4-3, and especially from two related systems, helps to convince on the biological relevance of the dimeric and tetrameric assembly, and the binding mode of the 5hmC present in both co-crystal structures. However, some of the structural data, and specifically the ligand bound structure do not convince me about the mechanism of action of the system determined with the biochemical data. Two major issues are present, and should be expanded upon. The first issue is the relatively weak and incomplete density of the ligand. The authors should present scoring from validation software or report metrics (such as bonding network and/or B-factors of ligand atoms and surrounding bound protein atoms) to allow the reader to decide on the density quality. Softwares such as 'Checkmyblob' would also help to verify the ligand density is interpreted correctly. The second issue is that the crystal structure contains only a single nucleotide rather than DNA incorporating the modified nucleotide. The issue with this is that the available surface in a DNA duplex containing the modified nucleotide is vastly different from that present in the single nucleotide. The authors suggest a similar binding mode in UHRF2-DNA structure, but more comparison and description is required to convince a reader this is the case.

The authors should consider the following corrections in order to strengthen the manuscript, and to present more clearly their findings.

Major Comments

The authors do not report the methods used to co-crystalise PD-T4-3 with 5hmC. Given the density presented in the paper (Fig 5a), it would be critical to know the details of crystallisation for these structures. More comparison between the apo and co-crystal structure would also be insightful

In the revised manuscript, we provide the crystallization conditions in the Methods part. We made a new figure for the comparison between the apo and co-crystal structures (Fig. S7). The structural comparisons between apo and ligand-bound states

demonstrate subtle conformational alterations in both the autoinhibitory loop and overall tetrameric assembly.

In the validation reports, the density for the O157:H7 PD-T4-3 5hmC looks to be of a better overall quality, covering more of the ligand. Could the authors show this density also in Fig 5a. The resolution of APEC O1 is higher than that of O157:H7, but the density for the ligand is much worse. Could the authors comment on this?

In the revised manuscript, we show the O157:H7 PD-T4-3 5hmC density in Fig. 5a. In the 5hmC-complexed structures of both APEC O1 and O157:H7, the electronic density of the 5hmC base is good while the triphosphate group has not enough density (Fig. 5a). In the structure of UHRF2-SRA in complex with a 5hmC-containing DNA, the 5hmC base is flipped out of the DNA duplex to bind UHRF2-SRA (Zhou et al. Molecular Cell 2014) (Fig. 5a). The recognition mode observed in our PD-T4-3/5hm-dCTP complex closely parallels this DNA-bound conformation. This observation is reasonable, since the recognition occurs between PD-T4-3 and the base of 5hmC, while the phosphate group is not involved. Thus, the unbound triphosphate group is likely flexible in the crystal structure.

Figures containing structures are difficult to interpret (e.g. fig4, fig5). Insets are busy and the slabbing such that much of the insets are almost transparent.

We have adjusted the depth of focus and remade the insets of Fig. 4 and Fig. 5.

Figure 4c insets - difficult to see details of insets due to small size and clipping such that many details are too transparent. Authors should enlarge to show more clearly the details.

We have modified Fig. 4c (Fig. 4b in the new figure 4) according to the reviewer's suggestions.

Figure 5a inset - show a second inset with the structures overlaid in order to show that the arrangement of the nucleotides are similar in each

structure.

According to the reviewer's suggestion, we made a new figure to show the similar arrangement of the nucleotides (Fig. 5b).

A figure showing the autoinhibitory loop position in the modified cytosine bound structure would be insightful, as well as a comparison of the apo and ligand bound structure. If no conformational change is seen, a description to this effect would be insightful. Is there any relative motion of units within the tetramer upon binding the ligand?

According to the reviewer's suggestion, we made a new figure (Fig. S7). The autoinhibitory loop is in the N-terminal GIY-YIG domain while the modified cytosine binds to the C-terminal domain. The structural comparisons between apo and ligand-bound states demonstrate subtle conformational alterations in both the autoinhibitory loop and overall tetrameric assembly. We propose that the ligand binding, instead induces conformational changes, prolongs the retention of the DNA substrate with the enzyme to outcompete the negatively-charged loop to approach the active site and get degraded (since DNA is also a highly negatively charged molecule).

Minor comments

The description of the additional tyrosine on ln140 is confusing. The authors explain the situation better in the discussion, and so the first reporting could be reworded for clarity as in the discussion.

In the revised manuscript, we have reworded this part for clarity.

Fig 6d – show the lack of secondary structure above sequence alignment as a line rather than gaps.

We have modified Fig. 6d as reviewer #2 suggested, as well as Fig. 3c.

Reviewer #3 (Remarks to the Author):

This paper from Liu et al examines a recently identified anti-phage defense system provisionally called PD-T4-3. The authors here show that the C-terminal domain of PD-T4-3 specifically recognizes phage DNA carrying hydroxymethylated cytosines with the N-terminal GIY-YIG domain then cleaving the DNA, making PD-T4-3 a variant of type IV restriction enzymes. Although the paper provides new insights into this anti-phage defense system, significant questions remain about the structural and mechanistic basis by which the system discriminates phage DNA from host DNA. These questions and other issues are articulated below:

Major questions/issues:

1) The data presented here suggest that PD-T4-3 defends against T2, T4, and T6 equally well. But in the original paper that identified the system, defense is far stronger against T4 than against T2 and T6. This discrepancy warrants some explanation. Is it because the PD-T4-3 homologs here are different or because the host strain being used is different? Using the original PD-T4-3 system as a control would help address this question.

It is not uncommon that different homologs of a bacterial defense system have different activity against phages. For example, different homologs of the CapRel system confer protection against different phages (Zhang, T. et al. Nature, 2022). A recent paper specially addresses this issue (Aframian et al. Nature Microbiology, 2025).

2) Is the recognition of both 5hmC and 5ghmC necessary for PD-T4-3 based defense? The gel shown in Fig 2d shows that both 5hmC- and 5ghmC-DNA are recognized, but what is the significance of this in vivo? It would be more convincing that these modifications are required for defense if the authors could show that deglycosylated strains of T-even phages are able to plaque better on PD-T4-3+ lawns. Additionally, is it possible to generate T-evens without 5hmCs as well (and then use a strain of E. coli without RM systems for plaquing)? If unmodified T-even phages are able to escape PD-T4-3, it would be more convincing that the modifications are necessary for PD-T4-3 based defense.

Since PD-T4-3 recognizes both 5hmC and 5ghmC, phages containing either 5hmC or 5ghmC will elicit the anti-phage activity of the PD-T4-3 system. We have sent e-mails

for four different groups asking for the T4 phage strains with deglycosylated 5hmC or unmodified cytosine, but have not yet got any response. Fortunately, I met Dr. Pan Tao at a conference in August and he kindly shares with us these two T4 phage strains: T4(hmC) phage (with deletion of both α -glucosyltransferase and β -glucosyltransferase) and T4(C) (with amber mutation in dCMP hydroxymethylase and deletion of dCTPase). As expected, deglycosylated T4 phage [T4(hmC)] is still sensitive to PD-T4-3 while T4(C) phage loses the sensitivity to PD-T4-3 (Fig. S4), confirming that the cytosine modification is necessary for PD-T4-3 based defense.

3) Related to the above point, it's a little unclear how the C-terminal domain is specifically sensing 5hmCs or 5ghmCs. In the structure shown in Fig 5a, the authors show that the main chain oxygen of Y127 interacts with the hydroxyl group of the modified cytosine. They then later make a point that since the modified cytosine is bound to the edge of the protein and not a binding pocket, the C-terminal domain can sense both 5hmC and 5ghmC (lines 249–251). Could PDT46 potentially bind promiscuously to other hydroxylated base modifications found in phage DNA, such as arabinosyl-hmC from RB69?

It is really an interesting question whether PD-T4-3 can recognize other cytosine modifications such as arabinosylated cytosine. We have sent e-mails asking for the phage strains with arabinosylated cytosine modifications (RB69, Bas46, and Bas47), but have not yet received a response. We notice that Bas46 and Bas47 are included in the BASEL collection (*PLoS Biology* 2021), therefore we contact Dr. Alexander Harms. Dr. Harms kindly replied to me saying that they need to generate new BASEL copies for shipping, but this may take 2-3 months since they have quite a backlog of earlier requests. We are eager to investigate the potential broader specificity for PD-T4-3 once the strains are accessible.

4) How PD-T4-3 specifically targets phage DNA while sparing host DNA is still unclear. PD-T4-3 features a negatively charged loop that occludes the active site, leading the authors to speculate that it may serve to prevent degradation of the host chromosome. The authors propose a model in which binding of the C-terminal domain to modified DNA is then followed by phage DNA 'outcompeting' the loop to enable degradation. But it could be that binding to modified DNA leads to an active movement of the loop to enable cutting of phage DNA. A structural analysis of PD-T4-3 bound to a longer

piece of DNA containing modified cytosines is probably needed to address this question and provide insight into how PD-T4-3 exclusively targets phage DNA. As it stands, the authors have only examined PD-T4-3 bound to 5hm-dCTP, which offers limited insight. The authors also might be able to test if a non-cleavable 5hmC DNA analog can prime cleavage of free DNA, which would support the allosteric model.

We thank reviewer #3 for pointing out this possibility. We have tried to obtain the structure of PD-T4-3 complexed with 5hmC-containing DNA. However, despite extensive efforts, we failed to get the complex crystals.

The ligand recognition site and the cleavage site reside in distinct domains. The structural comparisons between apo and ligand-bound states demonstrate subtle conformational alterations in both the autoinhibitory loop and overall tetrameric assembly (Fig. S7). Thus, we think it is unlikely for PD-T4-3 to operate through an allosteric mechanism.

We have not figured out how to obtain a non-cleavable 5hmC DNA analog. To check whether PD-T4-3 degrades DNA via an allosteric mechanism, we evaluate its endonuclease activity against ordinary DNA in the presence of 5hmC-modified DNA or 5hm-dCTP. Our data suggested that neither 5hmC-modified DNA nor 5hm-dCTP primed the cleavage of the ordinary DNA (Fig. S8).

5) The authors state that the GIY-YIG nuclease domain of PD-T4-3 and its homologs is novel because it has an extra tyrosine, but it's not clear why this tyrosine is significant or what role it plays. A mutation in that tyrosine does disrupt defense, but the function/role is unclear.

It has been proposed that a tyrosine in the active site of GIY-YIG nuclease acts as the general base that accepts a proton from the attacking water molecule and loses its OH proton in the process. However, whether the GIY tyrosine or the YIG tyrosine is the general base is still controversial (Truglio et al. The EMBO Journal 2005; Sokolowska et al. Nucleic Acids Research, 2011). In this manuscript, we show that PD-T4-3 has an extra tyrosine to form a "GIYxY-YIG" motif (Fig. 3c), and mutation of the extra tyrosine abolishes its anti-phage defense and endonuclease activity (Fig. 3d & 3e). The exact role of this third tyrosine needs further and carefully studies, which is beyond the scope of the current study.

6) The crystal structures reveal potential tetramerization, but the authors do not probe whether the interface formed between dimers or within a dimer are necessary for activity in vivo. Can mutations be designed based on the structure to test each interface and its relevance to defense? Also, can the oligomerization state of the protein be assessed by size exclusion chromatography to determine whether it forms a dimer, tetramer, both, or something else?

We thank reviewer #3 for the suggestions. Accordingly, we engineered mutations targeting distinct interfaces: the intra-dimer interface (R80A, V92S/L96S) and the inter-dimer interface (Δ loop β 1- β 2). The size exclusion chromatography profiles showed that Mutant R80A remained the same oligomeric state as the wild-type PD-T4-3, while the mutants V92S/L96S and Δ loop β 1- β 2 were eluted later than the wild-type PD-T4-3, indicating disassembly (Fig. 4f). Our data revealed a strong correlation between the oligomeric states and anti-phage activities (Fig. 4e), suggesting the tetrameric architecture of PD-T4-3 is required for its anti-phage activity. We also tried to obtain more accurate data of the oligomeric states of PD-T4-3 and its mutants, but the instrument of size exclusion chromatography/multi-angle scattering (SEC/MALS) analysis is not accessible for us.

Minor comments:

1) The authors should rename PD-T4-3. Those names were provisional – follow up studies on other systems identified in that original screen have since been renamed so PD-T4-3 should probably also get a better name that ideally reflects its function/activity. DefenseFinder should be updated accordingly.

We thank reviewer #3 for this constructive suggestion. We rename this system **CMoRE (Cytosine Modification Restriction Endonuclease)**. CMoRE recognizes the modified cytosine which is “MoRE than C”.

2) Line 54: it would be good to mention which E. coli strain PD-T4-3 was initially found in (ECOR68).

We have mentioned this information in the revised manuscript.

3) Lines 80–81: should indicate that these data confirm those already presented in Vassallo et al where defense as a function of MOI was reported for PD-T4-3.

We have indicated this information in the revised manuscript.

4) Line 97: it's mentioned that the phage DNA was severely depleted across the entire genome. This also supports the idea that this nuclease has limited sequence specificity, which would be consistent with the next few panels in Fig 2. Maybe they can mention this at this stage?

We thank reviewer #3 for this suggestion. We have mentioned this in the revised manuscript.

5) Line 98, 99: it's unclear how the divalent cation preference is relevant here. Maybe it makes more sense to talk about metal dependency prior to mentioning any *in vitro* assays, as the cation preference is presumably incorporated into those assays?

In the revised manuscript, we have moved the divalent cation preference results ahead, immediately after the first *in vitro* endonuclease assays (we need to determine the substrate first).

6) Line 113–123: the methodology described here was a bit opaque. It would be helpful to include a small diagram explaining the process. Additionally, in the corresponding Fig. 2f the N9 and N11 sites appear to be incorrectly positioned in the cartoon representation of the DNA.

We thank reviewer #3 for this constructive suggestion. We have added the diagram in Fig. 2f. For N9 and N11, we mean 9 nucleotides and 11 nucleotides, instead of the 9th and the 11th nucleotide.

7) Line 107–111: Can the EMSA assays be used to calculate KD? And if the authors performed an EMSA for 5ghmC DNA as well, maybe they can compare any differences in KD between these two substrates. In general, having KDs

would help to reveal the strength of the interactions reported. Also, why doesn't the DNA get cleaved in the EMSAs?

According to the reviewer's suggestion, we performed EMSA for 5ghmC DNA (Fig. 2e). It is very hard to calculate KD values based on the EMSA data, since EMSA is a semi-quantitative method and the smear on the gel makes it hard to be accurately quantified. The endonuclease activity of PD-T4-3 required divalent cation such as Mg^{2+} or Mn^{2+} (Fig. S3). We add 5mM EDTA when performing EMSA to avoid DNA cleavage (this information is provided in the Methods section).

8) Line 145: The finding that mutations in the catalytic residues reduce defense is compelling. It could be a nice correlation to show the same mutants are no longer capable of DNA cleavage using the previously described in vitro nuclease assay.

According to the reviewer's suggestion, we have purified these mutants and showed they lost the endonuclease activity (Fig. 3e).

9) Line 149: What's the length and sequence of the autoinhibitory loop? Would help to include it here.

In the revised manuscript, we have provided the information of the autoinhibitory loop, and the protein sequence of this loop has been shown in Fig. 6d.

10) Line 160: Is the bacterial toxicity studied using an inducible construct? There didn't seem to be a methods section for how bacterial spotting was done.

Yes, the bacterial toxicity assays use an IPTG-inducible construct. We have added this information in the methods section.

11) Line 160-162: The text suggests that this nuclease may adopt an inactive and active conformation. It would be helpful if the authors could elaborate on this point more explicitly and maybe speculate about the differences in the two conformations.

Actually, we think this nuclease always adopts an autoinhibitory conformation. Upon ligand binding, the recognition of the modified cytosine prolongs the retention of the DNA substrate within the enzyme, enabling the highly negatively charged DNA molecule to outcompete the autoinhibitory loop and gain access to the active site for degradation. In contrast, unmodified DNA molecules only transiently interact with the enzyme and unable to overcome the autoinhibition mechanism.

12) Line 189: Perhaps could explain what UHRF2 and an SRA domain is—was unsure of the significance.

In the structure of UHRF2-SRA in complex with a 5hmC-containing DNA, the 5hmC base is flipped out of the DNA duplex to bind UHRF2-SRA (Zhou et al. *Molecular Cell* 2014) (Fig. 5a). Structural comparison demonstrates that the recognition mechanism observed in our PD-T4-3/5hm-dCTP complex closely parallels this DNA-bound conformation. We reword this apart to clarify the significance.

13) Line 220, 221: how do these other PD-T4-3 systems with additional domains differ? Do they have altered specificity or other activities?

Our analysis shows that some PD-T4-3 systems contain additional domains (Fig. 6c). The TY-Chap2 domain is predicted to function as peptide-binding chaperone that target proteins modified by ADPr and potentially misfolded as a consequence. Most additional domains found in the PD-T4-3 systems is function-unknown. To elucidate the precise function of these additional domains needs long-term works, which is beyond the scope of the current study.

14) Line 278: This last claim about PD-T4-3 being a ‘reliable tool for 5hmC detection in mammals’ is an overstatement as there are no data about how PD-T4-3 interacts with eukaryotic DNA.

We agree that although PvuRts11, another restriction endonuclease recognizing 5hmC and 5ghmC, has been used as a tool to map 5hmC in mammalian genomes, there are no data for PD-T4-3 in this application. We replace “reliable” with “potential” to tone down.

15) Many sentences could be rewritten to improve clarity and there were a variety of grammatical issues throughout.

We thank reviewer #3 for pointing out this issue. We have double checked our manuscript and corrected the grammatical errors.

Comments on Figures:

Fig 1a: Should include the length of the gene. Or maybe protein product size/individual domain sizes?

We have added the protein boundaries for each individual domain in Fig. 1a.

Fig 2b: indicate what T4 gene is used for qPCR analysis (did not appear to be included in the methods).

For qPCR, we use a pair of primers specific to T4, not a specific gene. What really matters is the primer, not the amplified fragment. We have corrected the description in the figure legend and provided the primer sequence in the Methods part.

Fig 2c: is missing an x-axis label.

We have added the x-axis label.

Fig 2d, 2e: indicate what PCR product is being used for the nuclease activity assays and EMSAs. In the methods section the authors mention these fragments are amplified from the T4 genome, but it's unclear what the length of these fragments is and their cytosine/modified cytosine content. Same issue/question for Fig 3g.

In the revised manuscript, we have provided the information (length and cytosine content of the amplified PCR product) in the methods section.

Fig 3b. legend: “ α_2 - α_3 loops are highlighted by deep colors” should probably say “dark colors”.

We have corrected this issue according to the reviewer's suggestion.

REVIEWER COMMENTS

We highly appreciate the reviewers for their detailed and constructive comments that have significantly improved our manuscript. We have carefully revised our manuscript according to the reviewers' suggestions. In this revision, we have provided new data especially on the role and function of the unconventional tyrosine residue (Y19) within the 'GIYxY-YIG' motif (Fig. 3d & 3e) and the breadth of CMoRE activity (Fig. S10).

Reviewer #1 (Remarks to the Author):

The authors have improved their manuscript upon revision.

While I maintain that evaluating the breadth of CMoRE activity would certainly strengthen the paper – a view that is shared by Reviewer 3 – the authors explain that this exploration is not possible due to technical challenges or unavailability of biological material. Although this is a valid reason, I believe that phages Bas46 and Bas47 could be obtained from other labs – with Dr. HARMS permission – as the BASEL collection is now so widely distributed.

We thank Reviewer #1 for the suggestion. We have obtained the phages Bas46 and Bas47 with 5-arabinose-hydroxy-cytosine modification from Dr. Wenyuan Han with Dr. Harms' permission.

Both Bas46 and Bas47 contain double arabinosylation of hydroxy-cytosines (Fig. S9). In the revised manuscript, we have evaluated the anti-phage activity of CMoRE against Bas46 and Bas47 and the *in vitro* endonuclease activity of CMoRE against the genomic DNA of Bas46 and Bas47. Plaque formation assays revealed that CMoRE exhibited only very mild resistance to these phages (Fig. S10a). Consistent with this, CMoRE displayed significantly reduced endonuclease activity against the genomic DNA of Bas46 or Bas47 compared to 5ghmC-modified T4 genomic DNA (Fig. S10b). Collectively, our results indicate that CMoRE exhibits substrate recognition plasticity beyond 5hmC and 5ghmC, cleaving a spectrum of cytosine modifications with differential efficiencies.

Regarding my comment about normalisation bias in qPCR assays:

I apologize if my previous comment was unclear. I was not suggesting to co-

infect cells with T4 and T5 for the qPCR assay, which may introduce other problems. I was rather suggesting to perform qPCR assay on cells infected by T5 (not targeted by CMoRE) to monitor T5 DNA replication. The expectation with T5 is that phage DNA will increase over time, even in the presence of CMoRE. This is to show that variations in relative amount of phage DNA is not driven by host DNA depletion. Perhaps a better control would be to use a phage targeted by CMoRE but that is not inducing host DNA depletion but it is probably difficult to find a phage that matches these criteria.

That being said, looking now at the new panel in Fig 2c, the quantity of host DNA 8min post-infection is comparable in the two strain backgrounds which probably limits normalization bias I was expecting. In the light of this data, I would suggest that the authors restore their initial Fig2b.

We sincerely appreciate the valuable insights from Reviewer #1 regarding the consideration of normalization bias. According to the reviewer's suggestion, we restore the initial Fig2b.

My other comments have been satisfactorily addressed.

Reviewer #2 (Remarks to the Author):

The authors have made the necessary changes to the manuscript, and I am happy with the changes made.

We highly appreciate Reviewer #2 for the constructive comments that have significantly improved our manuscript.

Reviewer #3 (Remarks to the Author):

The responses and additional experiments from the authors have improved the clarity and strength of the paper. In particular, the authors now provide better support for the conclusion that T4 containing unmodified cytosines (deglycosylated T4(hmC)) is completely able to evade CMoRE. The in vivo experiments suggest that recognition of 5hmC alone is sufficient to trigger complete defense, which suggests that the variability in defense against

different T-even phages seen in some CMoRE homologues is likely attributable to factors other than glycosylation — perhaps differences in genome architecture or in the time scale of infection. The authors also now provide some additional data on a priming experiment to demonstrate that CMoRE does not employ an allosteric mechanism for DNA degradation. This helps explain better how CMoRE avoids autoimmunity. There are, however, some remaining issues with the manuscript:

1) As originally noted, the PD-T4-3/CMoRE system used here differs in its defense profile compared to that first published. In their responses the authors argue that this is because they are using a different homolog. I'm confused – is that really the case? Maybe I missed it, but it seemed initially like it was the same homolog/protein. If that's not the case, the authors should provide an alignment (unless I missed it) and describe the relationship of their homolog to the original one.

The PD-T4-3/CMoRE system in the first published paper is from ECOR68, one of the *E. coli* Reference (ECOR) set. The ECOR collection consists of 72 strains recovered from a range of hosts in different geographic locations. The collection was initially established to represent the diversity of the commensal *E. coli* population. The genomes for each strain have been sequenced and are available at: <https://doi.org/10.1128/mra.01133-18>

Below is the protein sequence alignment of PD-T4-3/CMoRE systems of ECOR68 and the three species in our study:

The sequence identity between ECOR68 and *E. coli* APEC O1 or *K. pneumoniae* is about 85%, and ECOR68 and *E. coli* O157:H7 share a sequence identity of 98.44%. Despite the high sequence identity, different homologs may have varying anti-phage activities. The similar phenomenon has been observed in other bacterial defense systems, such as the Metis system in a recent preprint of Dr. Rotem Sorek's group (bioRxiv preprint doi: <https://doi.org/10.1101/2025.11.05.686725>). The Metis systems from three highly similar *E. coli* strains (401675, 402837, and E308) exhibits varying levels of anti-phage activity. Please refer to below Fig. 1B from that preprint.

2) I still think the authors should test whether CMore can recognize arabinosylated cytosine. They indicate that they requested three such phages from Harms but don't yet have them. There are many other labs that have these phages and can likely provide them. Especially RB69, which isn't even part of the BASEL/Harms collection.

We thank Reviewer #3 for the suggestion. We have obtained the phages Bas46 and Bas47 with 5-arabinose-hydroxy-cytosine modification from Dr. Wenyuan Han with Dr. Harms' permission.

Both Bas46 and Bas47 contain double arabinosylation of hydroxy-cytosines (Fig. S9). In the revised manuscript, we have evaluated the anti-phage activity of CMore against Bas46 and Bas47 and the *in vitro* endonuclease activity of CMore against the genomic DNA of Bas46 and Bas47. Plaque formation assays revealed that CMore exhibited only very mild resistance to these phages (Fig. S10a). Consistent with this, CMore displayed significantly reduced endonuclease activity against the genomic DNA of Bas46 or Bas47 compared to 5ghmC-modified T4 genomic DNA (Fig. S10b). Collectively, our results indicate that CMore exhibits substrate recognition plasticity beyond 5hmC and 5ghmC, cleaving a spectrum of cytosine modifications with differential efficiencies.

3) The role of the additional tyrosine isn't elucidated. The authors argue it's 'beyond the scope of the current study' but this is noted even in the Abstract as one of the key features of this study ("Unique features, including a 'GIYxY-YIG' motif and an inhibitory negatively charged loop, distinguish CMore28 as a novel member of the GIY-YIG family.")

The unconventional 'GIYxY-YIG' motif is a unique feature for the bacterial CMore system. To investigate the functional role of the additional tyrosine, we performed site-directed mutagenesis and systematically evaluated both the *in vivo* anti-phage activity and the *in vitro* endonuclease activity of the resulting mutants.

Current evidence suggests that a tyrosine residue in the GIY-YIG nuclease active site may function as a general base, accepting a proton from the attacking water molecule (Truglio et al. The EMBO Journal 2005; Sokolowska et al. Nucleic Acids Research, 2011). We introduced mutations to the three tyrosine residues within the 'GIYxY-YIG' motif, generating two distinct mutant variants: tyrosine-to-phenylalanine (Y-to-F) and tyrosine-to-alanine (Y-to-A). The Y-to-F mutation specifically removes the hydroxyl

group from the tyrosine side chain, while the Y-to-A mutation eliminates both the hydroxyl group and the hydrophobic aromatic ring.

All mutations abolished the *in vitro* endonuclease activity of CMoRE (Fig. 3e). Interestingly, while mutations of the two conventional tyrosine residues (Y17 and Y31) to alanine or phenylalanine reduced anti-phage activity to a similar level, the Y19F mutant retained significantly higher activity than Y19A mutant (Fig. 3d, please refer to the figure below).

These results suggest that the conventional Y17 and Y31 likely function via their hydroxyl groups, whereas the additional tyrosine Y19 may rely on both its hydroxyl group and the hydrophobic aromatic ring for activity. We demonstrate that the additional tyrosine is really a unique feature.

4) It's interesting that some CMoRE catalytic mutants—despite completely abolishing endonuclease activity *in vitro*—did not seem to result in a very significant loss of defense *in vivo* (as shown in Fig. 3d and 3e). What is the explanation for this apparent discrepancy?

As shown in Fig. 3d, the mutations impaired the anti-phage function of CMoRE to varying degrees. We speculate that greater defense loss suggests a more critical role for the mutated site. It seemed that some mutants exhibited no significant loss of defense, likely due to the logarithmic scale of the Y-axis (Fig. 3d). For example, the mutant Y31A exhibited a 10-fold plaque-forming unit (PFU) increase versus the WT (Y31A: 3.3×10^5 vs WT: 3.3×10^4). This means that the mutant Y31A only retains 10% defense, which is a significant loss.

Although all the CMoRE catalytic mutants completely abolished the *in vitro* endonuclease activity (Fig. 3e), we speculate that they retain residual *in vivo* activity,

which may account for their anti-phage function (Fig. 3d). Since the residual activity is significantly reduced compared to the wild-type, the endonuclease activity of these mutants was undetectable under the current *in vitro* reaction conditions.

Minor comments:

In line 81, maybe the mechanism could be described as a “direct defense mechanism” rather than “phage-targeting” .

In the section describing the mutations used to disrupt oligomerization (lines 190 - 196), it might be helpful to clarify that the R180A mutant did not alter oligomerization (as shown in Fig. 4f), which is consistent with the lack of impact on anti-phage activity.

We have revised our manuscript according to the reviewer's suggestions.

REVIEWERS' COMMENTS

Reviewer #3 (Remarks to the Author):

The authors have made some additional changes and updates that improve the manuscript, which I support publication of at this stage.

We thank reviewer #3 for the positive comments.